



# Can GCMs represent cloud adjustments to aerosol–cloud interactions?

Johannes Mülmenstädt[1], Andrew S. Ackerman[2], Ann M. Fridlind[2], Meng Huang[1], Po-Lun Ma[1], Naser Mahfouz[1], Susanne E. Bauer[2], Susannah M. Burrows[1], Matthew W. Christensen[1], Sudhakar Dipu[3], Andrew Gettelman[1], L. Ruby Leung[1], Florian Tornow[2,4], Johannes Quaas[3], Adam C. Varble[1], Hailong Wang[1], Kai Zhang[1], and Youtong Zheng[5,6]

[1]Atmospheric, Climate and Earth Sciences Division, Pacific Northwest National Laboratory, Richland, WA, USA
[2]NASA Goddard Institute for Space Studies, New York, NY, USA
[3]Leipzig Institute for Meteorology, Leipzig University, Leipzig, Germany
[4]Columbia University Center for Climate System Research, New York, NY, USA
[5]Atmospheric and Oceanic Science Program, Princeton University, Princeton, NJ, USA
[6]Department of Earth and Atmospheric Science, University of Houston, Houston, TX, USA

**Correspondence:** J. Mülmenstädt (johannes.muelmenstaedt@pnnl.gov)

**Abstract.** General circulation models (GCMs), unlike other lines of evidence, indicate that anthropogenic aerosols cause a global-mean increase in cloud liquid water path ($\mathcal{L}$), and thus a negative adjustment to radiative forcing of the climate by aerosol–cloud interactions. In part 1 of this manuscript series, we showed that this is true even in models that reproduce the negative correlation observed in present-day internal variability of $\mathcal{L}$ and cloud droplet number concentration ($N_d$). We stud-
ied several possible confounding mechanisms that could explain the noncausal cloud–aerosol correlations in GCMs and that possibly contaminate observational estimates of radiative adjustments. Here, we perform single-column and full-atmosphere GCM experiments to investigate the causal model-physics mechanisms underlying the model radiative adjustment estimate. We find that both aerosol–cloud interaction mechanisms thought to be operating in real clouds – precipitation suppression and entrainment evaporation enhancement – are active in GCMs and behave qualitatively in agreement with physical process
understanding. However, the modeled entrainment enhancement has a negligible global-mean effect. This raises the question whether the GCM estimate is incorrect due to parametric or base-state representation errors, or whether the process understanding gleaned from a limited set of canonical cloud cases is insufficiently representative of the diversity of clouds in the real climate. Regardless, even at limited resolution, the GCM physics appears able to parameterize the small-scale microphysics–turbulence interplay responsible for the entrainment enhancement mechanism. We suggest ways to resolve tension between
current and future (storm-resolving) global modeling systems and other lines of evidence in synthesis climate projections.

## 1 Introduction

Increased aerosol concentration modifies cloud properties by increasing cloud droplet number, which initially makes clouds more reflective. When the aerosol concentration increase is due to an agent external to the climate system, for example, anthropogenic emissions, this cloud-brightening aerosol–cloud interaction (ACI) exerts a negative radiative forcing (RFaci) on





the climate. However, clouds then adjust to the cloud droplet number ($N_d$) perturbation by changing their liquid water path ($\mathcal{L}$) and cloud coverage; this enhancement or weakening of the instantaneous RFaci is called the radiative adjustment due to $\mathcal{L}$ (RA$_\mathcal{L}$) or cloud fraction (RA$_{f_c}$).

General circulation models (GCMs) have long disagreed with other lines of evidence on the sign of RA$_\mathcal{L}$, predicting that anthropogenic aerosols increase $\mathcal{L}$ when observational and large-eddy simulation (LES) estimates predict that $\mathcal{L}$ decreases
(Bellouin et al., 2020). Recently, Christensen et al. (2023), Varble et al. (2023), and Mülmenstädt et al. (2024) showed that several Coupled Model Intercomparison 6-generation GCMs (Eyring et al., 2016) produce negative correlations between cloud droplet number concentration $N_d$ and liquid water path $\mathcal{L}$ in present-day internal variability. This is welcome news, because the inability of GCMs to match observations was interpreted as GCMs' inability to represent enhanced cloud-top entrainment of dry air at high $N_d$. Enhanced entrainment is the dominant RA$_\mathcal{L}$ mechanism according to assessments based on multiple
lines of evidence.

However, even GCMs that produce negative $N_d$–$\mathcal{L}$ correlations in the present day still predict an $\mathcal{L}$ increase in response to anthropogenic aerosol emissions. In other words, the causal response of the model climate to secular changes in aerosols has the opposite sign of the correlation in present-day internal variability. This is concerning, for the negative correlation in observations is one pillar on which the sign of RA$_\mathcal{L}$ rests in assessments based on multiple lines of evidence. Part 1 of
this manuscript series (Mülmenstädt et al., 2024) offered several hypotheses for confounders that could produce a noncausal negative correlation between $N_d$ and $\mathcal{L}$.

In this manuscript, we return to entrainment-mediated evaporation of clouds as an adjustment mechanism and the question whether this mechanism is represented in GCMs. If so, then GCMs would also agree with the second pillar on which our multi-line assessment of RA$_\mathcal{L}$ rests: LES of cloud turbulence–microphysics interactions that shows a causal mechanism by which
increased droplet number results in increased entrainment drying of stratocumulus cloud (Ackerman et al., 2004; Bretherton et al., 2007). We will show that a broadly similar causal mechanism appears to exist in GCMs. This is also concerning, because the GCM results suggest that the well-understood and exhaustively LES-modeled RA$_\mathcal{L}$ in subtropical stratocumulus (Sc) clouds may not be representative of the global-mean RA$_\mathcal{L}$.

The results of Mülmenstädt et al. (2024) and this manuscript, taken together, cloud the Bellouin et al. (2020) picture of
reduced $\mathcal{L}$ in response to anthropogenic aerosol. It is possible that the known weaknesses of the tools at our disposal (observations, process modeling, and global modeling) are causing us to misunderstand the sign of RA$_\mathcal{L}$. We conclude with recommendations for using the complementary strengths of our tool set to increase the robustness of multiline assessments of ACI adjustments.

## 2 Data and methods

The hypothesis that our methods are designed to test is that a causal connection exists between $N_d$ and $\mathcal{L}$ in GCM physics that proceeds via enhanced cloud-top entrainment. Thus, we focus on causal and mechanism-denial experiments in single-column





and three-dimensional (3D) atmosphere runs to elucidate the causal link between $N_d$ and $\mathcal{L}$, and on diagnostics of cloud-top entrainment to ascertain that enhanced entrainment is involved in RA$_{\mathcal{L}}$.

## 2.1 Models

We use two of the three CMIP6-era models analyzed by Mülmenstädt et al. (2024) that produce an "inverted v"-shaped $N_d$–$\mathcal{L}$ correlation: the U.S. Department of Energy Exascale Earth System Model (E3SM) and NASA Goddard Institute for Space Studies (GISS) ModelE3. These models have different turbulence schemes; as entrainment-mediated ACI mechanisms must involve at least the turbulence and microphysics parameterizations, it was desirable to include model diversity in this study.

In part 1, we used E3SMv1; here we use E3SMv2 (Golaz et al., 2022) instead because it is significantly more efficient 60 at archiving the large, high-frequency fields required for the entrainment diagnostics. E3SMv2 differs from v1 largely in the parametric tuning (Ma et al., 2022) rather than in changes to the physics formulation. The $N_d$–$\mathcal{L}$ relationship documented in E3SMv1 persists in E3SMv2 for Sc clouds (Fig. S1).

In part 1, we used a ModelE3 parameter tuning derived through machine learning; here we use the default tuning from the ModelE3 development team, which produces a very similar $N_d$–$\mathcal{L}$ relationship but does not produce the oscillations in 65 surface precipitation at the lowest $N_d$ values that resulted from an assertive subgrid-scale multiplier tuning of autoconversion in nonturbulent layers.

## 2.2 Cloud selection

As the process understanding of entrainment-mediated RA$_{\mathcal{L}}$ is based chiefly on a small number of canonical subsidence Sc cases, our main focus is on understanding this cloud type in the GCMs, as well. Thus, while the eventual goal is to understand 70 the full spectrum of clouds that occur in the real atmosphere, for now we apply a restrictive set of criteria to maximize the similarity between model clouds and Sc:

– warm (cloud-top temperature warmer than freezing and zero ice water path), overcast (cloud fraction $f > 0.9$) columns,

– in locations where the dynamic–thermodynamic criteria of Medeiros and Stevens (2011) are met at least 30% of the time in the annual mean (see part 1),

– during Sc season (northeastern Pacific: Jun–Aug; southeastern Pacific and southeastern Atlantic: Oct–Feb),

– and with an inversion between model levels 10 and 15 from the surface (approximately 750–1400 m).

This reduces the complication that the cloud sample may comprise different cloud regimes governed by different ACI mechanisms (Mülmenstädt and Feingold, 2018), ensures validity of the cloud-top entrainment diagnostics, and avoids difficulties in the interpretation of cloud-top entrainment in partly cloudy model columns.



### 2.3 Single-column model experiments


We use an extensively studied, idealized subsidence Sc-like case specification to construct single-column experiments that are designed to investigate the mechanisms underlying the response of entrainment and $\mathcal{L}$ to $N_d$ in GCM physics.

#### 2.3.1 DYCOMS-II RF02 case description

The initial conditions and forcings for the single-column models (SCMs) follow the specifications of Ackerman et al. (2009)

for an intercomparison of lightly drizzling Sc. This setup is based on measurements during research flight 2 (RF02) of the second Dynamics and Chemistry of Marine Stratocumulus (DYCOMS-II) field study off the California coast (Stevens et al., 2003; vanZanten et al., 2005), obtained from horizontally averaging a notably heterogeneous field of somewhat heavily drizzling open cells within barely drizzling closed cells. With regard to the sensitivity of cloud thickness to entrainment in this case, the inversion is sufficiently strong and the overlying air sufficiently dry that it is not close to the "cloud deepening through entrain-

ment" regime of Randall (1984), and thus entrainment is expected to thin the cloud layer, as found in the LES intercomparison. [As an aside, we note that there is a sign error in equation (3) of Ackerman et al. (2009) specifying the total moisture profile above the inversion: the difference in the innermost brackets should be $z_i - z$ rather than $z - z_i$ as written.]

The Wyant et al. (2007) SCM intercomparison study used nearly the same setup as Ackerman et al. (2009), and both studies found that including drizzle and cloud droplet sedimentation generally slowed entrainment and enhanced domain-mean liquid

water path among a variety of models. Like the previous LES intercomparison of nocturnal Sc by Stevens (2005), the idealized setup ignored horizontal advective tendencies of cooling and drying associated with the large-scale flow for subtropical Sc decks as well as any solar radiation, consistent with the approximately 5-h aircraft sampling a nocturnal boundary layer air along an approximately Lagrangian trajectory. These and other simplifications such as constant subsidence and turbulent surface fluxes are consistent with the 6-hour simulation duration for the DYCOMS-II RF02 intercomparisons.

#### 2.3.2 GISS ModelE3 SCM description

The GISS ModelE3 SCM is a single-column version of the ModelE3 GCM that includes a number of updates to the column moist physics, as summarized in Cesana et al. (2021) and described in more detail by Cesana et al. (2019); unpublished manuscripts will document the model physics parameterizations more completely and also discuss the machine-learning approach to tuning the atmospheric model. In SCM mode the resolved advection is neglected and vertical advection is treated by

multiplying the vertical wind by the local gradient of all prognostic variables, as done for LES with periodic lateral boundary conditions (e.g., Stevens, 2005; Ackerman et al., 2009) to avoid complications associated with representing a divergent flow in a one-dimensional framework. The ModelE3 SCM allows for a number of specified forcings to override the native model parameterizations, which for this case consists of the following: (1) the radiative transfer uses a Beer's Law treatment that computes cloud-top cooling and cloud-base warming from the respective cumulative water paths downward from above and

upward from below, (2) geostrophic wind forcing is computed as in the LES framework using a fixed profile of geostrophic wind and the prescribed latitude, and (3) the surface drag is computed using a fixed friction speed.



For the ModelE3 SCM simulations here, we depart from the DYCOMS-II RF02 LES intercomparison specification in two ways. For the sake of simplicity, instead of a bimodal cloud condensation nuclei (CCN) distribution we specify a single lognormal mode of ammonium bisulfate with geometric mean radius 60 nm and geometric standard deviation 1.7. We also
extend the duration of the simulations to 24 h to check whether the clouds reach a steady state in each model; the latter half (hours 12.5–24) is not further analyzed.

Appendix A compares the SCM behavior against LES and assesses variations in the SCM setup that differ from those used for the E3SM SCM.

### 2.3.3 E3SM SCM description

The E3SM SCM is described in Bogenschutz et al. (2020). DYCOMS-II RF02 is part of the standard E3SM SCM case library. The main differences compared with the ModelE3 SCM are as follows. Since E3SMv2, the SCM uses the same vertical advection scheme as the three-dimensional model. The idealizations active in the baseline experiment are: prescribed surface heat fluxes, prescribed geostrophic wind, prescribed profile of divergence, and prescribed bimodal aerosol profile, as in the ModelE3 setup. Prescribing the surface wind stress or friction velocity is not supported in the E3SM SCM; during spinup, the
SCM stabilizes to $u^* \approx 0.4$ m s$^{-1}$, substantially higher than the DYCOMS-II RF02 case specification ($u^* = 0.25$ m s$^{-1}$).

Differences in model configuration in sensitivity experiments and $N_d$ susceptibility scans are described in the discussion of those experiments.

### 2.4 3D GCM configuration

The models analyzed in part 1 produced RA$_\mathcal{L} < 0$ in the default model configuration, presumably because of precipitation
suppression. To disentangle the opposing RA$_\mathcal{L}$ of precipitation suppression and a potential entrainment mechanism (RA$_\mathcal{L} > 0$), we turn off the precipitation suppression in E3SM by setting the exponent on $N_d$ in the autoconversion parameterization to zero, removing the explicit $N_d$-dependence of the autoconversion process. To maintain a climate state similar to the default model, we increase the autoconversion scale factor (Mahfouz et al., submitted). This is equivalent to presenting autoconversion with a globally constant $N_d \approx 50$ cm$^{-3}$ and results in present-day top-of-atmosphere flux and cloud radiative effect changes
< 1 W m$^{-2}$ compared with the default configuration. In the warm-cloud over-ocean mean, $\log \mathcal{L}_{PD} - \log \mathcal{L}_{PI} = 2.0 \times 10^{-3}$ in this model, indicating that switching off the precipitation suppression mechanism eliminates the strong negative RA$_\mathcal{L}$ of the default model configuration but does not expose a strong positive RA$_\mathcal{L}$ in its stead. The $N_d$–$\mathcal{L}$ correlation becomes more negative in Sc clouds when precipitation suppression is turned off (Fig. S2).

### 2.5 Entrainment diagnostics

As the causal ACI mechanism hypothesized to lead to RA$_\mathcal{L} > 0$ is enhanced entrainment with increasing $N_d$, we make it a focus of this paper to understand how entrainment behaves in the models. To this end, we use an entrainment diagnostic that



calculates the mixing between free troposphere (FT) and well-mixed boundary layer as a residual term in the mixed-layer budgets of water and temperature.

Let $q_v$ and $q_l$ be water vapor and liquid mixing ratio, $\theta$ and $T$ potential temperature and temperature. In adiabatic expansion and condensation, the total water mixing ratio

$$q_t = q_v + q_l \tag{1}$$

and liquid-water potential temperature $\theta_l$ are conserved; we approximate

$$\theta_l = \theta - \frac{L_v}{c_p} \frac{\theta}{T} q_l, \tag{2}$$

with $L_v$ the latent heat of evaporation of water (which, for simplicity, we take as temperature-independent, using its value at 273 K) and $c_p$ the isobaric specific heat of dry air. Budget equations for $\theta_l$, $q_t$, and total mass, vertically integrated over the planetary boundary layer (PBL), involve fluxes of water, dry air, and heat across the boundaries of the PBL (Lilly, 1968; Stevens, 2002; Caldwell et al., 2005; Kalmus et al., 2014; Mellado, 2017). Crucially for our purposes, this includes the entrainment flux into the boundary layer. We express the budget equations following Kalmus et al. (2014) but modify the notation to highlight the similarity with source and sink terms in a prognostic equation in Lagrangian form:

$$h \frac{\hat{D}(\rho \theta_l)}{\hat{D}t} = -\frac{\Delta F}{c_p} + \frac{L_v \Delta R}{c_p} + \frac{SH}{c_p} + E_\theta(\theta_l^+ - \hat{\theta}_l) \tag{3}$$

$$h \frac{\hat{D}(\rho q_t)}{\hat{D}t} = -\Delta R + \frac{LH}{L_v} + E_q(q_t^+ - \hat{q}_t) \tag{4}$$

$$\rho|_{z=h} \frac{\partial h}{\partial t} + \mathbf{v} \cdot \nabla_H h = E_h - \omega/g, \tag{5}$$

where $\rho$ is the density, $\Delta F$ the radiative cooling, $\Delta R$ the precipitation mass flux at the surface, LH and SH the latent and sensible heat fluxes at the surface, $h$ the PBL geometric depth, $\omega = Dp/Dt$ the large-scale pressure velocity, and $g$ the gravitational acceleration. The operator $\nabla_H$ is the horizontal gradient. Quantities with a + superscript (i.e., $q_t^+$ and $\theta_l^+$) are evaluated just above the inversion. Quantities with a caret ($\hat{A}$) or in angular brackets ($\langle A \rangle$) are mass-weighted vertical averages of a quantity $A$, evaluated at model-level midpoints $k$ between the lowermost atmosphere level $k_{\text{sfc}}$ and the uppermost level below the inversion $k_{\text{pbl}}$:

$$\hat{A} = \langle A \rangle = \frac{1}{\langle \rho \rangle h} \sum_{k=k_{\text{sfc}}}^{k_{\text{pbl}}} \rho_k \Delta z_k A_k; \tag{6}$$

the PBL-averaged material derivative of a 3D quantity $A$ is defined as

$$\frac{\hat{D}(\rho A)}{\hat{D}t} = \langle \rho \rangle \frac{\partial \langle A \rangle}{\partial t} + \langle \rho \mathbf{v} \cdot \nabla_H A \rangle + \langle \rho \rangle (A|_{z=h^-} - \langle A \rangle) \mathbf{v} \cdot \nabla_H h. \tag{7}$$

Each of the budgets of $\theta_l$, $q_t$, and $h$ (3)–(5) depends on an entrainment mass flux: $E_\theta$, $E_q$, and $E_h$, respectively. Physically,

$$E_\theta = E_q = E_h. \tag{8}$$



However, models do not necessarily respect this equality. Therefore, we retain the freedom to diagnose $E_\theta$, $E_q$, and $E_h$ sep-
arately; in the following, we use the degree of equality between these fluxes as a criterion for model fidelity to the physical
system.

## 3 Results

In Sect. 3.1, we describe the effective entrainment in the Sc regime in E3SM according to the entrainment diagnostics intro-
duced in Sect. 2.5. We analyze which properties of the atmospheric column influence the entrainment. To perform an unam-
biguous demonstration of a causal effect of increased aerosol on entrainment and PBL drying, we then turn to SCM analysis
in ModelE and E3SM in Sect. 3.2. We return to the 3D atmosphere in a model configuration without precipitation suppression
in Sect. 3.3 to search for evidence of a causal mechanism leading to reduced $\mathcal{L}$ in response to anthropogenic aerosols.

### 3.1 GCM effective entrainment

In a numerical model, the spatial discretization can potentially alter the behavior of the PBL in a qualitative way. The physical
Sc-topped PBL entrains free-tropospheric air by $O(1\,\mathrm{m})$-scale turbulent exchange through a sharp buoyancy barrier (Wood,
2012, and references therein). In the model, the static stability due to thermodynamic jumps across the inversion at PBL top is
less localized and weaker due to the finite vertical resolution. Depending on the model resolution, the resolved-scale advection
scheme, and the turbulence parameterization, vertical mixing across the poorly resolved inversion may be too strong because
the stability reported to the turbulence scheme is underestimated or because fluctuations in the resolved-scale vertical velocity
mix the airmasses instead of moving the boundary between them, that is, lead to "numerical diffusion". Collectively, we term
these behaviors "model artifacts". The problem with model artifacts in mixing is that the effect of such mixing on the PBL
temperature and humidity need not have the correct susceptibility to the host of anthropogenic perturbations – forcing by and
adjustments to both aerosol and greenhouse-gas forcings, as well as feedback mechanisms in response to anthropogenic global
warming – that are hypothesized to influence cloud-top entrainment by changing the atmospheric state ($N_d$, temperature and
humidity in the PBL and FT, and FT emissivity).

Therefore, our task is to determine whether the mixing in the model behaves more like physical entrainment or more like
artificial mixing. If we calculate $E_\theta$, $E_q$, and $E_h$ in (3)–(5) as residuals, then they describe the mixing between PBL and FT,
including both the entrainment and model artifacts. We then apply three criteria that help us make that determination:

1. In the real atmosphere, $E_\theta = E_q = E_h$ all describe the same entrainment mass flux that comes about due to turbulent
processes at the boundary-layer top. In a numerical model, however, equality of the entrainment fluxes is not a given.
  For one thing, models treat $\theta_l$ and $q_t$ differently, for example, to ensure nonnegative-definite $q_t$. For another, the length
  scales at which entrainment occurs reach below 1 m, far beyond the resolved dynamics of most types of models. Mix-
  ing between the boundary layer and FT in a model, therefore, results from a combination of resolved advection and
  parameterizations. Having multiple independent measures of the entrainment mass flux affords us the ability to ask both
whether the model-diagnosed entrainment estimates are consistent and whether they are physical. Consistent fluxes are



highly correlated, with $E_q$, $E_\theta$, and $E_h$ close to a 1:1 regression slope. (In an Eulerian model, $E_h$ is difficult to diagnose when the advective tendency of $h$ over a model time step is small compared to the vertical resolution, which is the case in GCMs. We restrict our analysis to $E_q$ and $E_\theta$.)

2. Even if the diagnosed fluxes are consistent, however, they can still be unphysical. That is, the mass flux could be detraining air out of the boundary layer instead of entraining into the boundary layer.

3. Finally, the dependence of the entrainment flux on atmospheric conditions can indicate that the wrong mechanisms are at work in the model. For example, a strong dependence of entrainment on the FT vertical velocity would indicate overly strong vertical advection through the capping inversion.

Thus, we propose three measures of the realism of the entrainment representation in a model: the joint distribution of $E_\theta$ and $E_q$, the sign of the mass flux, and the dependence of the entrainment flux on the atmospheric state.

Under Sc conditions (as defined in Sec. 2.2), E3SM produces vertical profiles of $q_t$ and $\theta_l$ consistent with a fairly well-mixed PBL capped by a fairly sharp thermodynamic jump. Figure 1 shows composite vertical profiles stratified by PBL depth.

The effective entrainment qualitatively agrees very well with physical understanding of the Sc-topped PBL. The fluxes derived from the separate budgets agree well with each other, yielding a close relationship with slope near 1 (see joint probability in Fig. 2). Furthermore, the sign of the fluxes is consistent with physical entrainment from the FT into the PBL ($E > 0$; see the marginal cumulative distribution functions in Fig. 2) rather than showing a distribution including both positive and negative values, which would be consistent with numerical diffusion. The instantaneous entrainment also qualitatively responds in the expected way to instantaneous variability (as opposed to climatological spatial variability, seasonal temporal variability, etc.) in properties of the atmospheric column (Fig. 3). Entrainment increases with surface heat fluxes and cloud-top radiative cooling, consistent with increased turbulence production leading to increased entrainment; decreases with the magnitude of the thermodynamic jumps at the inversion, consistent with a stronger buoyancy barrier suppressing entrainment (and the moisture jump being strongly correlated with the temperature jump); and is independent of the instantaneous grid-scale vertical velocity, consistent with large-scale subsidence moving the boundary between airmasses (i.e., the FT and the PBL) rather than mixing them.

In summary, the entrainment behavior of the GCM, at least qualitatively, appears largely free of numerical artifacts due to the coarse model resolution. We can, therefore, focus instead on the effects of anthropogenic perturbations on the modeled entrainment and on how the parameterized model physics affects those entrainment responses.

## 3.2 Entrainment ACI mechanism in single-column runs

Figure 3 shows that entrainment depends on numerous properties of the atmospheric column. This creates a bewildering web of possible causal and covariability effects by which entrainment internal variability could be correlated with aerosol internal variability in 3D atmosphere runs. SCM runs, in contrast, provide a clean way to diagnose cause and effect in GCM column physics, which is a reason they are widely used during model development. First, the SCM allows us to hold any combination of boundary conditions on a single column fixed. Thus, we can switch off any effects mediated by the grid-scale horizontal





circulation. These effects includes synoptic-scale confounders of the type discussed by Mülmenstädt et al. (2024). Second,

and relatedly, we are free to explore the effects of model physics on ACI without having to retune the model to global-mean energy balance. Thus, we avoid the difficult problem whether to attribute changes in model behavior to the physics changes under investigation versus the nuisance changes required to restore energy balance that might also affect the ACI behavior (e.g., Golaz et al., 2011; Mülmenstädt et al., 2020, 2021).

A welcome side effect (and a raison d'être) of SCM use is that the experiment setup closely matches the LES runs that inform

so much of our process understanding. Like Ackerman et al. (2004), Bretherton et al. (2007), and Hoffmann et al. (2020), we can focus on well-understood subtropical subsidence-region Sc and vary one boundary condition – the aerosol concentration – at a time (and, if desired, one model-physics mechanism at a time). This does not address the question whether the model results are representative of the global-mean effective radiative forcing – that is best addressed with global runs – but it answers the question whether LES and GCM column physics respond similarly to perturbations around an as-near-as-possible identical

base state.

The E3SM SCM uses the two-mode prescribed aerosol concentration profile specified for RF02 (Wyant et al., 2007), while the ModelE3 SCM uses a single accumulation mode aerosol size distribution as described above. We then modify the amplitude of this profile to elucidate the causal effect of $N_d$ change on $\mathcal{L}$. In the ModelE3 SCM, we scan the aerosol number concentration $N_a = \{20, 30, 40, 60, 80, 120, 160, 320, 640\}$ cm$^{-3}$. In the E3SM SCM, we scale the prescribed aerosol concentration up and

down by a factor of 8: $N_a = N_{a_0} \times \{1/8, 1/4, 1/2, 1, 2, 4, 8\}$. Prescribing aerosol eliminates ACI mechanisms in which clouds affect the aerosol state, such as potential effects of aerosol scavenging on the $N_d$–$\mathcal{L}$ relationship (McCoy et al., 2020). This simplifies the attribution of $\mathcal{L}$ responses to $N_d$ perturbations by removing one class of processes from consideration.

Figures 4 and 5 show the ModelE3 and E3SM SCM time series. In both models, the PBL deepens, indicating entrainment in excess of the subsidence rate; the PBL in both models also deepen a similar amount when native longwave radiative transfer

is used (see Appendix). With the exception of a short duration before and after steps in the discretized PBL depth in E3SM, both models maintain an overcast cloud; the loss of cloud cover in E3SM's SCM when the PBL top jumps by a model level is clearly a model artifact, so these periods are excluded from further analysis by requiring $f > 0.9$, as in the 3D model analysis both here and in part 1. Furthermore, the discrete PBL depth increases are associated with discontinuities in the entrainment diagnostics: a dependence of $E$ on how long the top model level of the PBL has been subject to entrainment in E3SM, and a

reversion to a constant $E$ once the PBL has deepened in ModelE3. We may be mitigating the E3SM artifact by averaging over two full deepening cycles, effectively averaging over the dependence of $E$ on position in the deepening cycle. We attempt to mitigate the ModelE3 artifact by only averaging $E$ until the first PBL deepening occurs.

As expected, varying the aerosol concentration strongly affects the droplet concentration. From this response, the two SCMs then diverge in the details of their behavior, but they reach the same behavioral endpoint at sufficiently large $N_d$: enhanced

entrainment leading to a loss of $\mathcal{L}$ as $N_d$ increases. In ModelE3, $\mathcal{L}$ after spinup starts out with a monotonically increasing $N_d$ dependence; after spinup, the low-$N_d$ runs experience an increase in $\mathcal{L}$, while the high-$N_d$ runs experience a decrease, leading to a time-average $\mathcal{L}$ that first increases with $N_d$ and then decreases (Figs. 4 and 6). The increasingly negative $\mathcal{L}$ tendency as a function of $N_d$ accompanies an increasingly strong entrainment warming and drying (Fig. 6). The increase in entrainment with





increasing $N_d$ competes with a decrease in precipitation (Fig. 4). ModelE3 with native longwave radiation, however, maintains

three times greater $\mathcal{L}$ after 24 h duration (see Appendix).

In E3SM, $\mathcal{L}$ after spinup has a monotonically decreasing relationship with $N_d$, which remains true at each point in time throughout the runs. As all runs experience fairly rapid $\mathcal{L}$ loss with time, time averages only show weak dependence on $N_d$ (Fig. S3). We can instead quantify susceptibilities by scanning across the different aerosol experiments at each time step; this is shown for $\mathcal{L}$ in Fig. 7 and for $E$ in Fig. 8. Precipitation in E3SM largely ceases after the first time step, even though drizzle

was measured in the DYCOMS-II RF02 observations. In ModelE3 with native LW radiation, precipitation experiences sharp peaks with a periodicity similar to PBL depth increases.

Whether the effect of varying the aerosol concentration on $\mathcal{L}$ is expected depends on our Bayesian prior. If our expectation for the $\mathcal{L}$ response is based on the RA$_{\mathcal{L}}$ results of Mülmenstädt et al. (2024), we would predict the causal effect of increased $N_d$ to be an increase in $\mathcal{L}$. If our expectation is based on the LES-based process understanding, then we would predict the

causal effect of $N_d$ to be a decrease in $\mathcal{L}$. The surprising result is that the SCM sides with the LES, not the response of the 3D GCM with which the SCM shares its model physics.

While the SCM behavior is consistent with our mechanistic understanding of entrainment-mediated drying, we need to point out several caveats. First, the details of what "entrainment-mediated drying" entails are different in the two models, as discussed above, and we will find in Sec. 3.3 that the behavior in the 3D E3SM run is more consistent with the ModelE3

SCM than with the E3SM SCM. Second, the E3SM SCM entrainment fluxes reach the equivalent of several centimeters per second entrainment velocity, significantly stronger than LES or the ModelE3 SCM produce for this case, and stronger than the E3SM 3D run produces for Sc on average. Third, if the E3SM physics had not serendipitously produced very low precipitation rates, the decrease in $\mathcal{L}$ with increasing $N_d$ would probably have been overwhelmed by the precipitation suppression signal. None of these caveats negates the finding that the GCM physics appears capable of producing entrainment-mediated $\mathcal{L}$ loss

qualitatively consistent with LES findings, and significant intermodel diversity is to be expected in SCM studies (Zhu et al., 2005; Wyant et al., 2007). They do, however, indicate that there is ample further process investigation to be performed in future work.

We conduct several additional E3SM SCM experiments with perturbed physics. These experiments further test that the causal effect of aerosols on $\mathcal{L}$ in SCM mode not only has the same sign as in LES but proceeds via the same physical mechanisms

and show that the entrainment-mediated drying can be tuned to agree quantitatively with LES.

**Sedimentation–entrainment feedback is the source of entrainment enhancement**  Size-dependent sedimentation is one of the processes by which higher-$N_d$ clouds lose liquid relative to lower-$N_d$ clouds in LES (Bretherton et al., 2007) under sufficiently dry overlying air (Ackerman et al., 2004). The Gettelman et al. (2015) microphysics parameterizes size-dependent sedimentation. Guo et al. (2011) showed through process denial experiments that $\mathcal{L}$ loss only occurs when this

process is included in the SCM version of the Geophysical Fluid Dynamics Laboratory (GFDL) AM3 model. Figure 9 shows that $\partial \log \mathcal{L} / \partial \log N_d \approx 0$ in the E3SM SCM as well when we switch off the sedimentation flux (and thus its size dependence).





**Parameter tuning can move the GCMs toward quantitative agreement with LES on susceptibility**  The ModelE3 SCM closely

replicates LES of the RF02 case (Appendix A). While the E3SM SCM behaves significantly differently than LES in that

it is nonprecipitating for most of the run, it does appear that its entrainment-mediated $\mathcal{L}$ susceptibility can be moved

closer to LES estimates by appropriate parameter choices. Figure 9 also shows that increasing the size-dependent sed-

imentation by a factor of 2 increases $\partial \log \mathcal{L}/\partial \log N_d$. Quantitatively, this brings the E3SM SCM closer to quantitative

agreement with LES ($-0.35 \le \partial \log \mathcal{L}/\partial \log N_d \le -0.22$; Ackerman et al., 2004, supplementary table 1). It may be pos-

sible to achieve quantitative agreement by combining the sedimentation tuning factor with similar tuning factors in the

turbulence parameterization. This would not be an outlandish model tuning, considering that it may be taking the role of

an "enhancement factor" (Covert et al., 2022) compensating for the coarse vertical discretization $O(100 \text{ m})$ compared to

the process scale $O(1 \text{ m})$.

**The liquid water path reduction could also have an important shortwave absorption component**  The DYCOMS-II RF02

SCM specification calls for a nocturnal simulation, i.e., without shortwave radiative effects. In the E3SM SCM, we per-

form a sensitivity test with shortwave radiative effects in which the sun rises at 12 h (not shown); this simulation is

less entraining overall, deepening only once, and is only able to sustain its cloud cover for $\approx 18$ h. During the daytime

portion (hours 12–18), this configuration's $\partial \log \mathcal{L}/\partial \log N_d$ is more negative than the LW-only simulation's. Absorption

of shortwave at cloud top exerts a warming effect that increases with $N_d$ (Stephens, 1978; Hoffmann et al., 2020). The

behavior of the SCM run with shortwave radiation is consistent with this $N_d$-dependent cloud-top heating. Cloud-top

heating counteracts longwave cloud-top cooling, reducing the entrainment. Shortwave heating increases with increasing

$N_d$, decreasing $\mathcal{L}$.

## 3.3 Entrainment ACI mechanism in 3D model runs

If the entrainment-mediated adjustment of $\mathcal{L}$ is evident in the SCMs, what becomes of it in the full 3D model atmosphere?

From Mülmenstädt et al. (2024), we know that RA$_{\mathcal{L}} < 0$, that is, $\mathcal{L}$ under present-day (PD) emissions is greater than $\mathcal{L}$ under

preindustrial (PI) emissions, opposite in sign to what is expected from the entrainment-mediated mechanisms and from the

SCM. To understand why this happens, we use diagnostics targeted at entrainment mechanisms and perform model experiments

designed to isolate entrainment mechanisms.

### 3.3.1 Indications of entrainment mechanisms in present-day correlations

Figure 10a shows cloud-top entrainment into the Sc PBL as a function of $N_d$ and $\mathcal{L}$. Two things are readily apparent in this

figure. First, entrainment has a strong dependence on $\mathcal{L}$. Entrainment is expected to increase with $\mathcal{L}$ based on the ability of

the cloud to generate turbulence, which depends on the availability of liquid water for evaporation and cloud-top radiative

cooling. The $\mathcal{L}$ values at which entrainment turns on are in reasonable agreement with recent LES (Hoffmann et al., 2020) and

observational (Zhang et al., 2022) results. Second, at a given $\mathcal{L}$, entrainment increases with $N_d$. This is shown quantitatively





in Fig. 11: in all three Sc regions, the entrainment susceptibility $\partial \log E / \partial \log N_d|_{\mathcal{L}} > 0$ except at low $\mathcal{L}$, and the cloud water
loss increases (the Eulerian tendency $\partial \mathcal{L} / \partial t$ becomes more negative) with increasing $N_d$.

The main conclusion from these plots is that the model produces greater entrainment in response to higher $N_d$ in Sc clouds
with strong entrainment. In other words, there appears to be mechanistic agreement between the model physics and process
understanding of RA$_{\mathcal{L}}$ via entrainment enhanced by increased droplet number. Further support for this conclusion comes from
the instantaneous $\mathcal{L}$ tendency $\partial \mathcal{L} / \partial t$. The regression slope $\partial^2 \mathcal{L} / \partial N_d \partial t|_{\mathcal{L}}$ is predominantly negative except at low $\mathcal{L}$. In other
words, clouds with a positive entrainment susceptibility also exhibit a negative liquid-water tendency, the magnitude of which
increases with $N_d$, confirming that there is a relationship between entrainment susceptibility and cloud water loss (presumably
to drying).

There is reason to be cautious, however. We have tested whether the known negative $N_d$–$\mathcal{L}$ correlation occurs in conjunction
with positive $N_d$–$E$ and negative $N_d$–$\partial \mathcal{L} / \partial t$ correlations, consistent with an entrainment drying adjustment to an $N_d$ increase.
This peels away one layer of confounding between $N_d$ and $\mathcal{L}$, increasing our confidence that there is a mechanistic link between
$N_d$, $E$, and $\mathcal{L}$. However, it is possible that the relationships of $E$ and $\partial \mathcal{L} / \partial t$ with $N_d$ are themselves confounded, just as the
regression between $N_d$ and $\mathcal{L}$ was found to be a result of covariability by Mülmenstädt et al. (2024).

Quantitatively, the regression entrainment susceptibility reaches values several times greater than the causal susceptibility
in the DYCOMS-II E3SM SCM runs. At the same time, the entrainment flux itself is closer to expected subsidence Sc values
than the DYCOMS-II E3SM SCM values. In both of these respects, the E3SM 3D atmosphere behaves more like the ModelE3
SCM than like the E3SM SCM. Furthermore, the entrainment susceptibility in the 3D runs is only weakly sensitive to scaling
the size-dependent sedimentation (not shown), unlike in the E3SM SCM. Possible explanations for these differences between
the E3SM 3D and SCM behaviors are that the correlation is confounded (see above) or that processes other than the size-
dependent sedimentation examined in the SCM also contribute to the entrainment enhancement or the cloud liquid loss. There
is no shortage of further model experimentation (e.g., done as part of a process-denial PPE) that could be done to shed more
light on the entrainment behavior of the model.

### 3.3.2   Absence of entrainment-mediated adjustment in PI and PD emissions experiments

In the Sc regime, there are hints of behavior in accordance with process understanding (e.g., Randall, 1984; Ackerman et al.,
2004): $\mathcal{L}$ decreases in response to the anthropogenic $N_d$ increase when the relative humidity (RH) in the FT (diagnosed from
the first model level above the inversion) is lowest, as shown in Fig. 12. However, the results come with multiple caveats. First,
the dependence on FT RH is not robust across Sc regions. While northeast Pacific (NEP) and southeast Pacific (SEP) Sc regions
show negative $\mathcal{L}$ susceptibility to anthropogenic $N_d$ at low FT RH and positive $\mathcal{L}$ susceptibility at high FT RH (with $\mathcal{L}$ decrease
overall), the southeast Atlantic (SEA) Sc region shows no clear pattern. Furthermore, the NEP and SEP behavior requires us
to select only the completely overcast (cloud fraction $f = 1$, rather than the default $f > 0.9$ requirement we use elsewhere; this
reduces the data sample from approximately $3.7 \times 10^5$ to $1.2 \times 10^5$ columns). Otherwise, the $\mathcal{L}$ susceptibility has no clear sign
or FT RH dependence across regions. A final puzzling observation is that the entrainment mass flux, entrainment temperature
flux, and entrainment moisture flux are all virtually unchanged between PD and PI emissions. On balance, the conservative





interpretation of these results is that any potential $\mathcal{L}$ reduction signal in response to anthropogenic aerosol is small enough to require far longer model runs.

In the global mean, $\mathcal{L}$ is virtually unchanged when emissions are changed from PI to PD in our E3SM configuration with deactivated precipitation suppression. (The reader may recall from Sect. 2.4 that the global-mean warm, overcast cloud $\Delta \log \mathcal{L} = 2.0 \times 10^{-3}$.) Thus, even if an entrainment-drying ACI mechanism is represented in the model (as the evidence from the SCM experiments in Sect. 3.2 and the PD statistics in Sect. 3.3.1 suggests), the model considers that mechanism's global effect to be negligible.

One possible explanation why the positive susceptibility of $E$ to $N_d$ seen in PD internal variability does not lead to a decrease in $\mathcal{L}$ is that entrainment susceptibility may beget its own demise (Zhu et al., 2005; Wood, 2012). (We reiterate, as throughout, that another possible explanation for relationships seen in internal variability is confounding.) It is true that $E$ appears to increase with $N_d$ at a given $\mathcal{L}$ (Fig. 10a). However, $\mathcal{L}$ is not fixed during the temporal evolution of a cloud; increased entrainment leads to loss of $\mathcal{L}$. But at lower $\mathcal{L}$, entrainment is weaker. Eventually, entrainment may even decrease $\mathcal{L}$ to a low

enough value that the cloud is protected from further entrainment drying (Hoffmann et al., 2020; Zhang et al., 2022). Cloud aggregate statistics are consistent with this interpretation, where the $E$ susceptibility to $N_d$ at fixed $\mathcal{L}$ is positive (Fig. 10a), but the overall susceptibility of $E$ on $N_d$ still becomes negative at sufficiently high $N_d$ (Fig. 10b) due to the strong negative correlation between $\mathcal{L}$ and $N_d$. If such a negative feedback mechanism is at play, it would be an example of buffering in the cloud system (Stevens and Feingold, 2009): an initial cloud loss process being shut off by the change in cloud state due to that

process.

## 4    Discussion, conclusions, and recommendations

We have documented two surprising behaviors from GCMs. The first is that GCMs can produce negative $N_d$–$\mathcal{L}$ PD correlations that do not predict $\mathrm{RA}_{\mathcal{L}}$ (Mülmenstädt et al., 2024). In terms of mechanistic understanding, the simplest explanation is that the correlation is due to confounding rather than a causal relationship involving the entrainment-mediated mechanisms suggested

by process scale modeling.

    This is where models had a second surprise in store: there is actually a causal negative relationship between aerosol and $\mathcal{L}$. The evidence for this causal relationship comes from SCM studies, where, like Guo et al. (2011), we find that increased aerosol, while holding all other boundary conditions fixed, leads to liquid-water loss. Furthermore, this loss appears to be due to entrainment, or at least it occurs in conjunction with increased entrainment when the aerosol boundary condition is increased. At

face value, this would seem to indicate excellent mechanistic agreement with LES-based process understanding that enhanced entrainment drying reduces $\mathcal{L}$. Three-dimensional atmosphere runs, too, show evidence for entrainment-mediated liquid-water loss in correlations between entrainment and $N_d$.

    However, like Karset et al. (2020), we find that a secular $N_d$ increase caused by anthropogenic emissions leads, at best, to a very weak decrease in $\mathcal{L}$, unlike what would be expected from the SCM idealized case study or the relationships found in PD

internal variability (increased $E$, increasingly negative $\partial \mathcal{L}/\partial t$ when $N_d$ increases). This may be a manifestation of buffering of





the cloud system against perturbations; in this case, the buffering mechanism is that enhanced entrainment leads to sufficient liquid-water loss to shut off entrainment driven by cloud-top radiative cooling, protecting the clouds from further liquid loss.

Summarizing the findings from parts 1 and 2 of this manuscript series, we come to the following conclusions. First, negative relationships between $N_d$ and $\mathcal{L}$ observed in PD internal variability are not necessarily indicative of a causal reduction in $\mathcal{L}$, and thus not necessarily predictive of decreased $\mathcal{L}$ when $N_d$ increases due to anthropogenic emissions. Second, causal negative relationships between $N_d$ and $\mathcal{L}$ in LES are not necessarily representative of the more diverse ensemble of clouds in the global-mean RA$_\mathcal{L}$. Thus, the disagreement on the sign of RA$_\mathcal{L}$ between global models and other lines of evidence (Bellouin et al., 2020) may not be solely due to a deficiency in the GCM physics; it could also be due to known deficiencies in the other lines of evidence.

Casting doubt on whether we even know the sign of RA$_\mathcal{L}$ is a highly unsatisfactory state of affairs. Answering the following six questions would provide a potential remedy:

**What complexity is required?** Many different processes are at play, and it is not clear which ones are represented in the models studied here, either through parameterization or emerging from the interplay of physics and dynamics. There is certainly value in (and models are suited to) studying how the climate response depends on the processes included the model. But trying to include all known or hypothesized processes could fall into the trap of amassing a zoo of "$n$th indirect effects" (Stevens and Feingold, 2009; Mülmenstädt and Feingold, 2018). Instead of overelaborating the model with redundant and competing parameterizations (Proske et al., 2023), the approach leading to the lowest climate projection uncertainty may lie in finding the minimal set of parameterizations that allow the model to reproduce physical process understanding of the sensitivities that matter for the climate problem: sensitivities to those boundary conditions that change with aerosol and greenhouse-gas ERF or with global warming. Evident qualitative differences in the E3SM and ModelE3 SCM behaviors compared with one LES and the challenges of well-constraining LES with observations are reminders that the representation of basic microphysical and turbulent processes still afford ample opportunity for tighter constraint.

**What resolution is required?** GCM resolution can offer, at best, a cartoon version of the mechanisms at play in real clouds. But "cartoon" does not have to be a pejorative; it is the simplest representation of reality that can convey the author's intent (wit, satire, heuristic simplification, or, when applied metaphorically to models, predictive skill for a different climate state). As noted in the previous paragraph, there is value in simplicity. There is a trade-off between resolution and simplicity, however: the coarser the resolution, the greater the reliance on the parameterized physics, and the longer the list of phenomena (e.g., mesoscale circulations within a GCM column) that need to be parameterized. How far this trade-off can be pushed determines the minimal resolution required for reliable climate projections. Terai et al. (2020) and other studies already provide hints at the answer. An important additional piece of information that can be obtained from the entrainment diagnostics presented here is how entrainment behavior changes, qualitatively and quantitatively, as model resolution is coarsened from LES (or, ideally, direct numerical simulation of the cloud-top turbulence; Mellado et al., 2018) to km-scale "storm-resolving" global models to 10–100 km-scale GCMs.





**How do base-state and process errors affect modeled climate responses?** Our results show that entrainment and its susceptibility are strong functions of $\mathcal{L}$ and $N_d$. The climate response may also be a function of the model's FT RH, which appears biased high in E3SMv2. Dependence on base state (Christensen et al., 2023; Varble et al., 2023) and competing processes (Mülmenstädt et al., 2020, 2021) in models necessitates careful evaluation of the base state $N_d$ and $\mathcal{L}$, paying close attention to issues of definition and aggregation (Elsaesser et al., 2017; Feingold et al., 2022; Varble et al., 2023).

Better constraints on the base state alone can run into "equifinality" (von Bertalanffy, 1950; Beven and Freer, 2001; Lee et al., 2016; Regayre et al., 2018; Mülmenstädt and Feingold, 2018) problems that negate a direct reduction in climate projection uncertainty (Lee et al., 2016; Regayre et al., 2018; Mülmenstädt et al., 2020, 2021; Zelinka et al., 2022). In the case of entrainment, however, the apparent strong dependence of the process representation on base-state errors may yield a significant payoff in tighter constraints on climate projections when the base state is improved.

**What observational constraints on the entrainment process are available?** Along with vital advances in teasing causality out of observations of cloud $N_d$ and $\mathcal{L}$ (Fons et al., 2023), the biggest step forward along the observational track would be better constraints on the entrainment process itself. One possibility may be to use subadiabaticity (Merk et al., 2016; Varble et al., 2023) as an indicator of the cloud liquid loss. This would only provide a time-integrated measure of the loss processes, and as such would require disentangling entrainment drying from precipitation.

**How representative are susceptibilities derived in small ensembles of individual cases?** Given the difficulty of placing observational constraints on entrainment, the most convincing evidence for entrainment-mediated RA$_{\mathcal{L}}$ continues to come from LES studies. Large ensembles of LES cases (e.g., Gustafson Jr. et al., 2020; Glassmeier et al., 2019) are vital to provide resilience against the possibility that the well-studied, often idealized canonical subsidence Sc conditions may not be representative of the global-mean role that cloud-top entrainment plays in ERFaci. These LES ensembles will 455 be particularly valuable if they span the initial-value and boundary-value problem aspects of the climate response (i.e., sample the vast variability of meteorology encountered by Sc clouds in the climate), and if they provide cloud lifecycle evolution (Kazil et al., 2021) that can be validated against cloud lifecycle observations (Christensen et al., 2020) sufficiently to ensure LES adequacy for purpose, given differing results in multi-LES studies (e.g., Ackerman et al., 2009). The same point on the importance of large ensembles holds for SCM studies: the differences between E3SM 460 and ModelE SCM of the DYCOMS-II RF02 case, as well as the differences between the high-entrainment E3SM SCM and moderate-entrainment E3SM 3D runs, illustrate the need for a set of SCM test cases that better approximate the diversity of meteorological conditions encountered in the climate. A way forward would be to perform the suite of SCM causal aerosol perturbation experiments and mechanism denial experiments from Sect. 3.2 on an ensemble of single-column cloud cases from a 3D run using the SCM ability to "replay" the forcing of the column by the 3D atmosphere 465 (Bogenschutz et al., 2020).

**Was precipitation suppression the bigger problem all along?** According to our results, and consistent with Karset et al. (2020), cloud-top entrainment, even when represented in the model physics, only appears to play a small role in the





global-mean RA$_\mathcal{L}$. If this GCM finding reflects reality, focusing on the precipitation-mediated component of RA$_\mathcal{L}$ takes on renewed importance.

Recognizing that the ACI climate problem is at its core a multiscale physics problem is crucial, as is recognizing that no single line of evidence is capable of putting our knowledge of RA$_\mathcal{L}$ on solid footing (Mülmenstädt and Feingold, 2018). The above research questions are a sketch of a multiscale modeling and observations roadmap. Simultaneously, by accounting for the multiscale nature of the problem, they would put us on a path of reliable climate projections beyond the global energy budget (e.g., projecting regional hydrologic extremes due to the spatial heterogeneity of ERFaci) by ensuring that global

modeling systems correctly represent both the spatial pattern of ERF and the response of the circulation at all scales to this forcing (Mülmenstädt and Wilcox, 2021).

**Appendix A: Tracing DYCOMS-II RF02 model behavior from LES to SCM**

The SCM setup here is based on the Ackerman et al. (2009) intercomparison of LESs, which is in turn based on airborne observations of a nocturnal marine Sc deck during the DYCOMS-II project (Stevens et al., 2003; vanZanten et al., 2005)

following an approximate Lagrangian trajectory over 5 hours (to paraphrase the description of Wyant et al., 2007). The purpose of this appendix is to connect LES of the case to the SCM setups and results in this study.

A representative LES in that study was the Distributed Hydrodynamic Aerosol and Radiative Modeling Application (DHARMA) model, here run with two-moment cloud microphysics (Tornow et al., 2021), and with a vertical grid spacing of $\delta z = 5$ m to 200 m above the original inversion to avoid a positive feedback between entrainment and grid spacing that arises on the Ack-

erman et al. (2009) specified grid, which was designed to accomodate simulations of duration 6 h instead of the 24 h used here.

The ModelE3 SCM is run here following the specifications of Ackerman et al. (2009), with one departure being that aerosol are treated as a monomodal lognormal distribution with a fixed number mixing ratio (corresponding to 60 cm$^{-3}$ at 900 hPa and 10°C) instead of the bimodal size distribution specified by Ackerman et al. (2009), which produces comparable $N_d$ values to

the LES, as seen in Fig. A1.

The LES and ModelE3 SCM model setups also both depart from the Ackerman et al. (2009) specificaton of cloud-water sedimention and instead use the treatment in their (similar) native microphysics schemes of cloud water sedimentation [assuming a gamma distribution with a relative dispersion of 0.3, per Geoffroy et al. (2010) in the SCM, and with a relative dispersion of 0.2 in the LES].

Given that this case was used for the development and default tuning of the ModelE3 SCM, it is not a surprise that the SCM results match the LES reasonably well, with the greatest differences being (1) modestly slower entrainment and, thus, deepening of the marine boundary layer (MBL) in the SCM, and (2) about a factor of two less drizzle reaching the surface for most of the duration. While the formulations of the SCM and LES are different, and such differences are to be expected, we note that a narrowing of the assumed droplet size distribution to match that in the LES has little impact and does not deepen

the MBL more over the 24-h duration (not shown). We also note that the stronger drizzle for the LES is not explained by its



assumption of a narrower raindrop size distribution, which on its own would instead be expected to result in weaker drizzle at the surface.

As briefly noted in the main text, the SCM setup used for the E3SM further departs from the Ackerman et al. (2009) specification and the ModelE3 SCM setup here in a number of ways, among them: (1) it did not use the specified surface stress, (2) it did not adopt the specified Beer's Law parameterization of longwave flux divergence, and (3) it did not apply the local subsidence rate to vertical gradients using first-order upwinding to avoid complications with divergent flow, but instead treated the specified divergence using the dynamic core. While we are unable to even begin to match departure (3), we are able to consider departures (1) and (2) with the ModelE3 SCM. For (1), we adopted the equilibrated surface stress from the E3SM results, which has has very little impact on the results (not shown). For (2) we used the ModelE3 native longwave radiation scheme, using the ModelE3 SCM standard machinery to patch in the McClatchey (1972) standard atmosphere above the 1.5-km top of the initial sounding provided by Ackerman et al. (2009). As seen in Fig. A1, doing so results in appreciably faster entrainment, which better matches the E3SM SCM results (Fig. 2), which also deepens by about 350 m over the 24-h duration.

The inverted-v relationship between $\mathcal{L}$ and $N_d$ is evident in the LES and ModelE3 SCM results (Fig. A2). The match to the LES results is not a surprise for the default tuning, as such a comparison was considered in the ModelE3 model development phase. However, the decent match of Tun1, a product of the machine-learning tuning of the parent GCM, results partly from skill, in limiting the range of parameters that the machine learning explored, and partly from luck, as some parameter combinations devised by the machine learning did not result in such a good match to the LES result in these terms (not shown).

*Code and data availability.* Following acceptance, the analysis code and model output will be released with a code and data DOI

*Author contributions.* All authors contributed to the experiment design, model runs, data analysis, or manuscript writing.

*Competing interests.* At least one of the (co-)authors is a member of the editorial board of Atmospheric Chemistry and Physics.

*Acknowledgements.* Bjorn Stevens's suggestion to use budget equations to characterize models' entrainment behavior provided the spark for this work. We thank Christopher Bretherton, Yao-Sheng Chen, Leo Donner, Graham Feingold, Tom Goren, Ed Gryspeerdt, Peter Kalmus, Jan Kazil, Adrian Lock, Roger Marchand, Daniel McCoy, Isabel McCoy, Roberto Mechoso, Brian Medeiros, Juan-Pedro Mellado, Yi Ming, Prasanth Prabhakaran, Phil Rasch, Christina Sackmann, Yunpeng Shan, Philip Stier, Shuaiqi Tang, João Teixeira, Chris Terai, Yoko Tsushima, Hui Wan, Rob Wood, Heng Xiao, Tak Yamaguchi, Mark Zelinka, Jianhao Zhang, and Xiaoli Zhou for comments and discussion. This work arises from the 2021 U.S. Climate Modeling Summit held virtually and co-chaired by Susanne Bauer and Gokhan Danabasoglu. JM was supported by Office of Science, U.S. Department of Energy (DOE) Biological and Environmental Research as part of the Earth System Model Development (ESMD) program area and used resources of the National Energy Research Scientific Computing Center (NERSC), a



U.S. DOE Office of Science User Facility located at Lawrence Berkeley National Laboratory, operated under contract DE-AC02-05CH11231.
The entrainment diagnostics were developed under the EAGLES project funded by the U.S. DOE ESMD and Atmospheric Science Research program areas. ASA, AMF, FT and SEB were supported by the NASA Modeling, Analysis, and Prediction Program and their computational resources were provided by the NASA Center for Climate Simulation (NCCS) at Goddard Space Flight Center. The Pacific Northwest National Laboratory (PNNL) is operated for DOE by Battelle Memorial Institute under contract DE-AC05-76RLO1830.



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



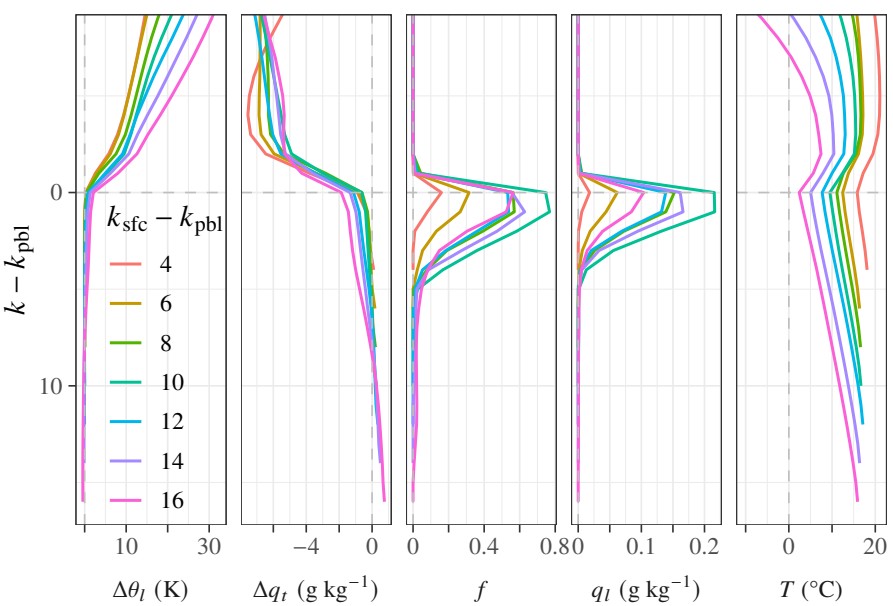

**Figure 1.** Composite vertical profiles in E3SM Sc conditions as defined in Sect. 2.2. (To enable plotting of a cloud cover profile, the $f > 0.9$ requirement is not applied.) The lowest level in the column at which the temperature increases with altitude is identified as the PBL top, with corresponding level number $k_{\mathrm{pbl}}$. The vertical coordinate is model level referenced to PBL top, $k - k_{\mathrm{pbl}}$ (positive downward). Profiles are stratified by PBL depth, measured as the difference between the PBL-top model level and the lowermost model level $k_{\mathrm{sfc}}$. (In E3SMv2, $k_{\mathrm{sfc}} = 72$.) Profiles of $q_t$ and $\theta_l$ are shown as differences $\Delta q_t$ and $\Delta \theta_l$ with respect to the mass-weighted vertical mean over the PBL.

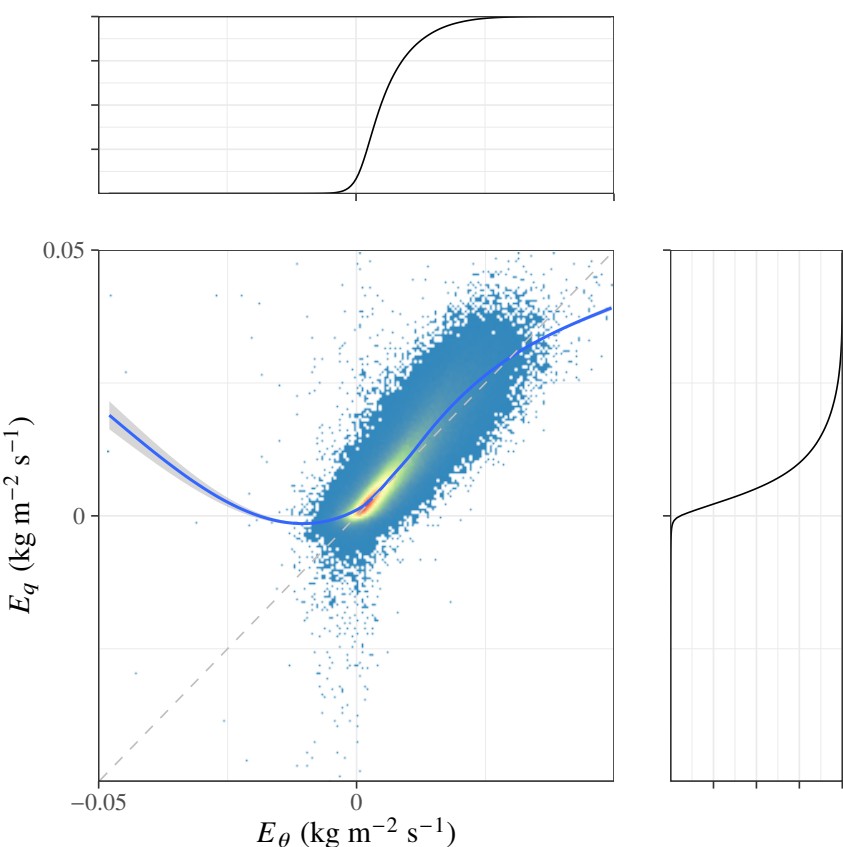

**Figure 2.** Physical consistency checks on $E_q$ and $E_\theta$. The central panel shows the joint probability density $P(E_q, E_\theta)$, along with a LOESS-smoothed mean $E_q$ as a function of $E_\theta$ (blue line) and dashed gray 1:1 line. The outer panels show the marginal cumulative distributions of $E_q$ and $E_\theta$.





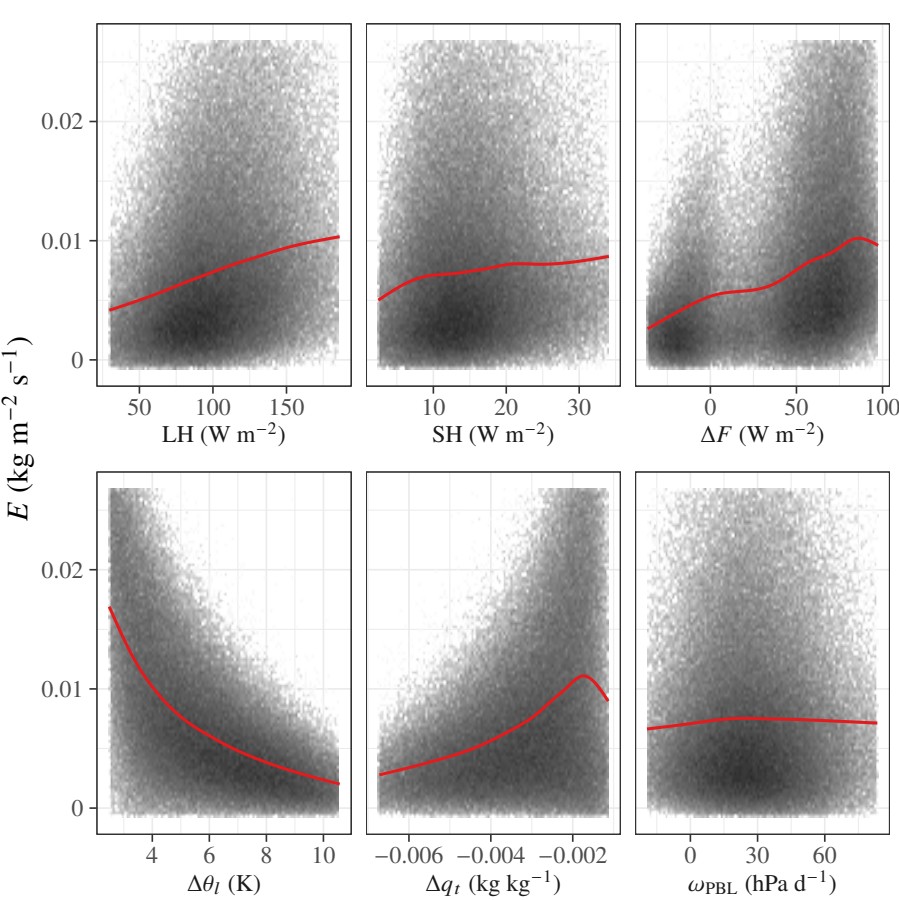

**Figure 3.** Dependence of entrainment on column properties: surface latent and sensible heat flux LH and SH, radiative cooling $\Delta F$ (cooling is positive), thermodynamic jumps across the inversion $\Delta\theta_l$ and $\Delta q_t$, and pressure vertical velocity at PBL top $\omega_{\text{PBL}}$.

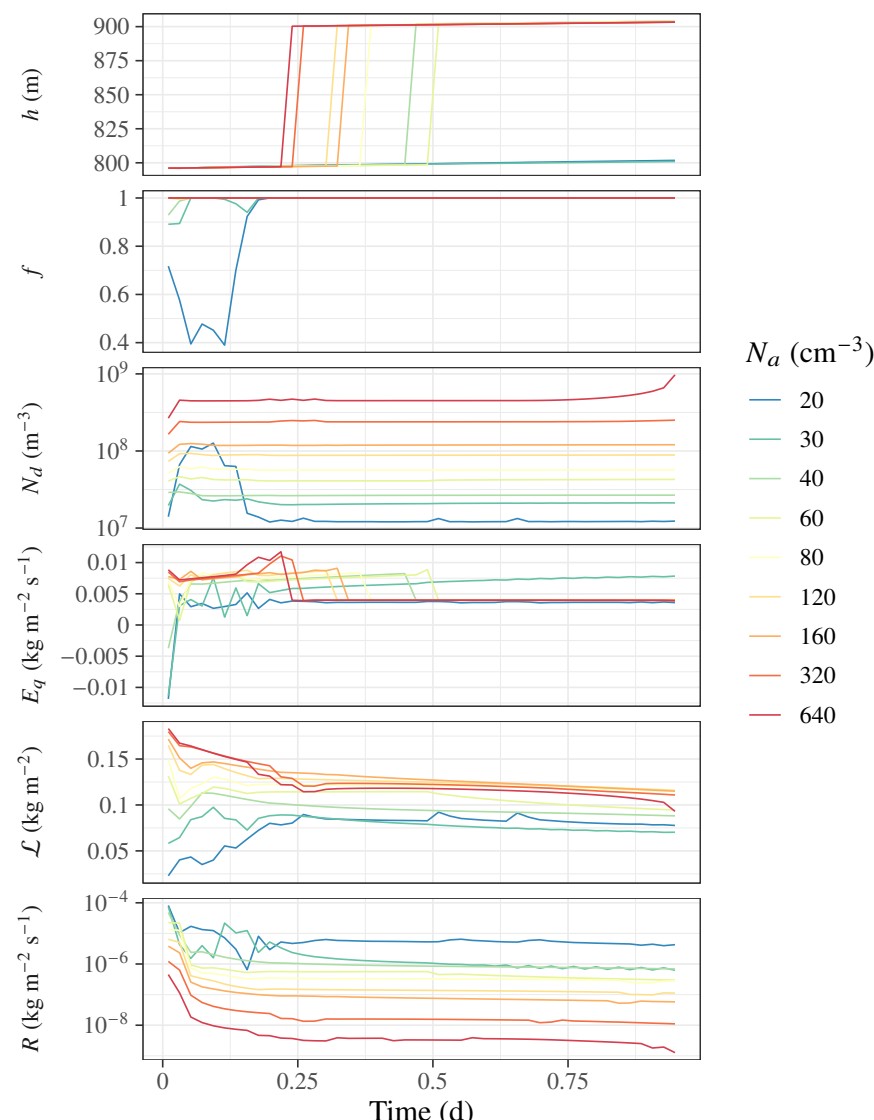

**Figure 4.** GISS ModelE3 SCM time series of PBL height (height of lowest model level with inverted temperature lapse) $h$, cloud cover $f$, mean droplet number $N_d$, entrainment $E$, liquid water path $\mathcal{L}$, and surface precipitation rate $R$ for the DYCOMS-II RF02 experiment. Prescribed aerosol concentration is varied between $N_a = 20$ and $640$ cm$^{-3}$.



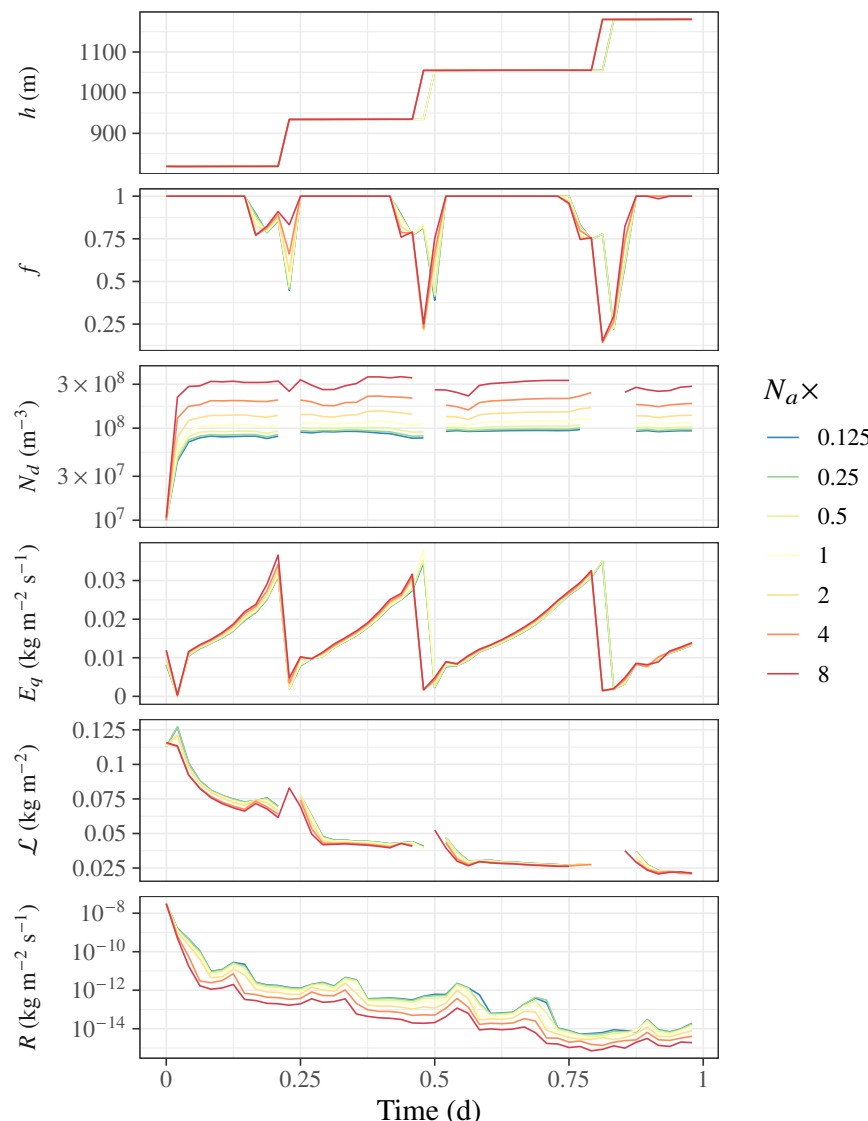

**Figure 5.** E3SM SCM time series for the DYCOMS-II RF02 setup. As in Fig. 4. Prescribed aerosol concentration is varied by a factor of 8 above and below its default value.





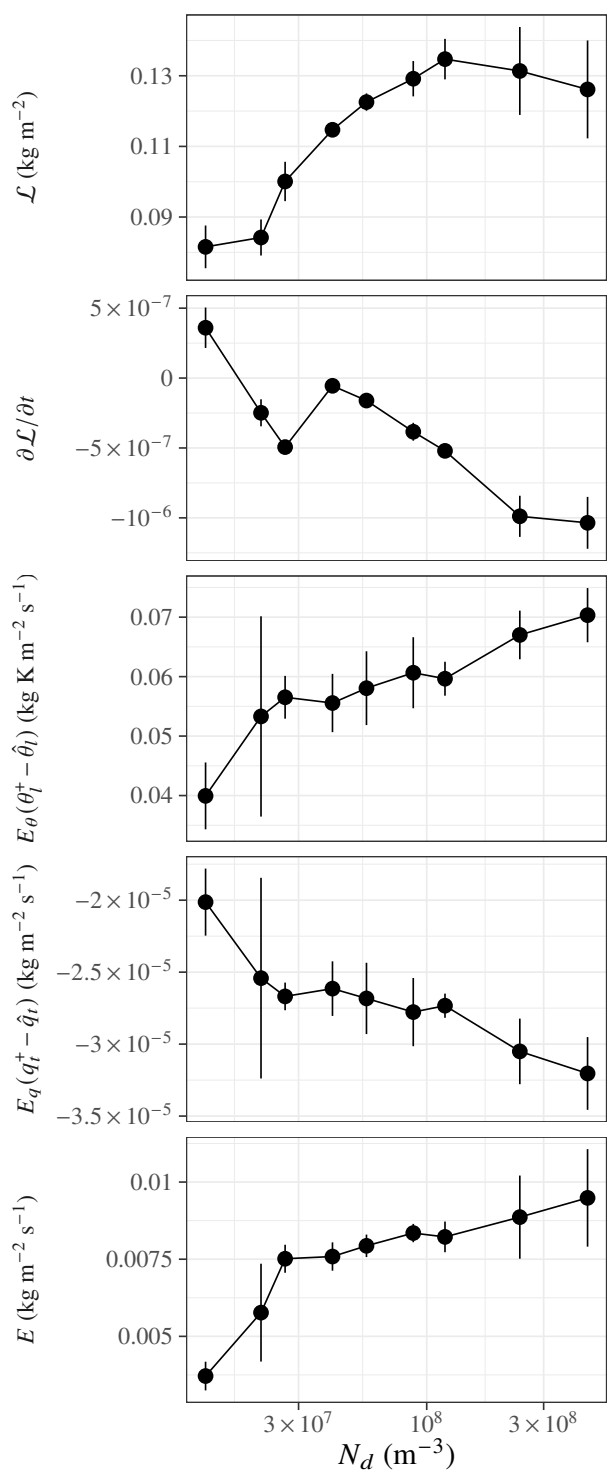

**Figure 6.** GISS SCM $N_d$ relationships. State variables are averaged over the time period from 2–12 h; error bars indicate the standard deviation. The tendency $\partial\mathcal{L}/\partial t$ is calculated by linear regression over overcast conditions ($f > 0.9$) between the end of spinup (2 h) and 12 h; error bars indicate the standard error on the regression slope. **29**

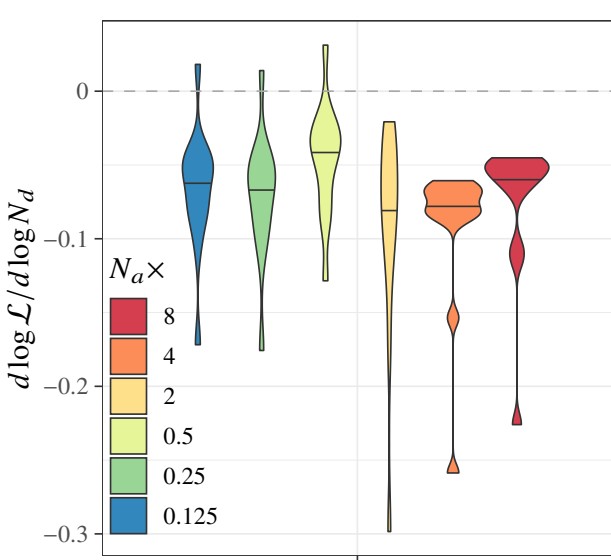

**Figure 7.** E3SM SCM $\partial \log \mathcal{L}/\partial \log N_d$. Susceptibilities are calculated at every time step from the $\delta \log \mathcal{L}$ and $\delta \log N_d$ in the perturbed-aerosol experiment relative to the default-aerosol experiment. The plot shows the density distribution of $\partial \log \mathcal{L}/\partial \log N_d$ over all time steps from the end of the spinup period (2 h) to 12 h, excluding periods when the cloud fraction drops below 0.9. Horizontal black lines across the density plots indicate the median.





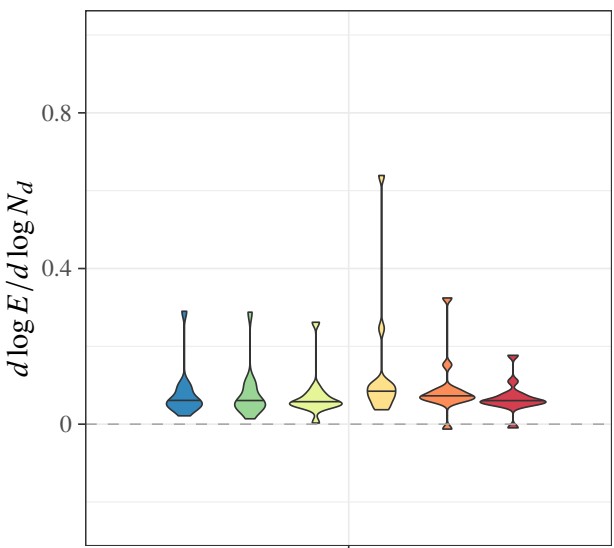

**Figure 8.** E3SM SCM $\partial \log E / \partial \log N_d$. As in Fig. 7 but showing the susceptibility of cloud-top entrainment.

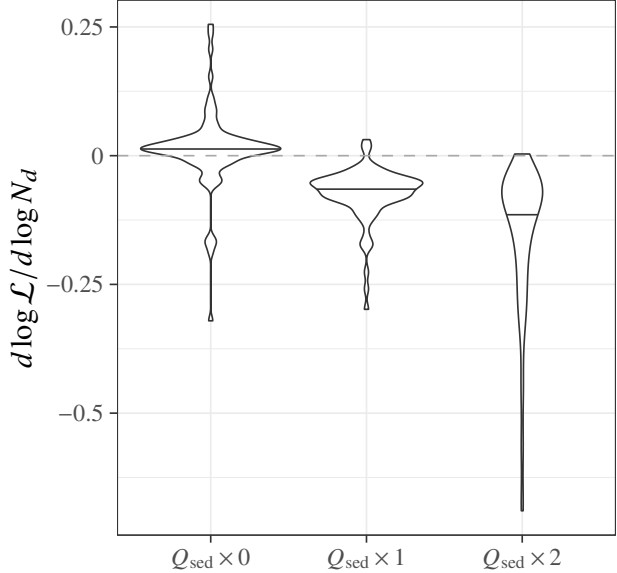

**Figure 9.** E3SM SCM $\partial \log \mathcal{L} / \partial \log N_d$ variation with size-dependent sedimentation. As in Fig. 7 but varying a scale factor applied to the sedimentation flux $Q_{\mathrm{sed}}$.

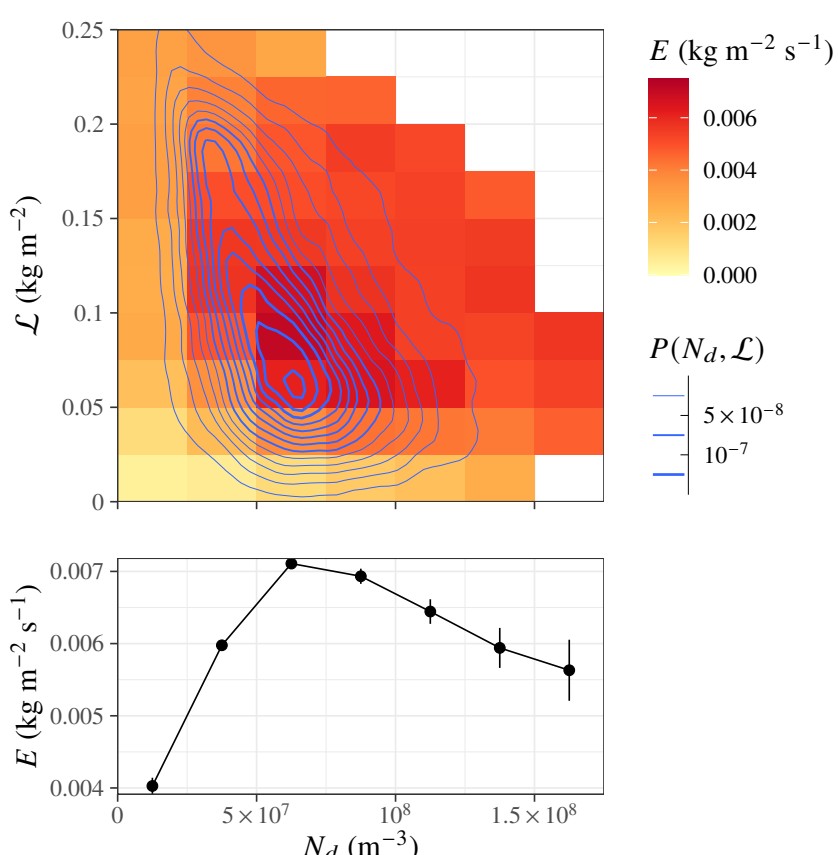

**Figure 10.** E3SM 3D atmosphere entrainment $E$. The top panel shows the dependence on $\mathcal{L}$ and $N_d$; only boxes with $n > 25$ points are included. Contours of the density $P(N_d, \mathcal{L})$ are overlaid. The bottom panel shows the dependence of $E$ on $N_d$ when the $\mathcal{L}$-dependent entrainment $E(N_d, \mathcal{L})$ is integrated over the $\mathcal{L}$ distribution; error bars indicate the standard error.



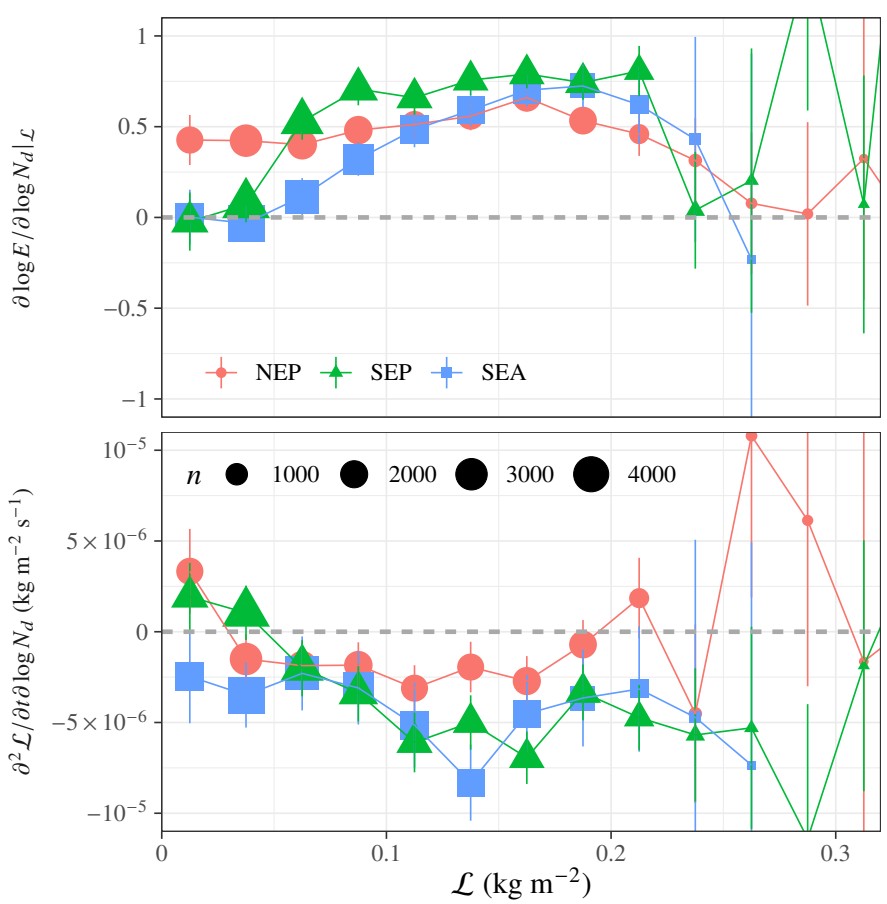

**Figure 11.** Susceptibility of entrainment and $\partial\mathcal{L}/\partial t$ to $N_d$. Susceptibility to $N_d$ is calculated as linear regression slope of $\log E$ and $\partial\mathcal{L}/\partial t$ against $\log N_d$ over instantaneous PD statistics within each $\mathcal{L}$ bin.





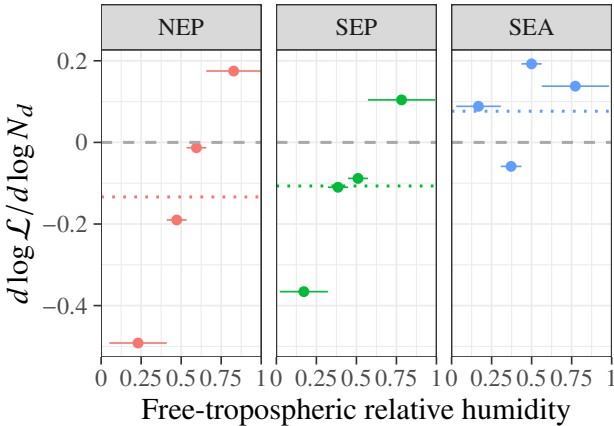

**Figure 12.** E3SM 3D atmosphere $d\log\mathcal{L}/d\log N_d$ stratified by free-tropospheric relative humidity quartiles. The susceptibility is calculated from the differences $\Delta\log\mathcal{L}$ and $\Delta\log N_d$ between PD and PI emissions runs averaged over each RH bin in each Sc region. Dashed lines indicate the regional Sc $d\log\mathcal{L}/d\log N_d$ mean integrated over free-tropospheric relative humidity.

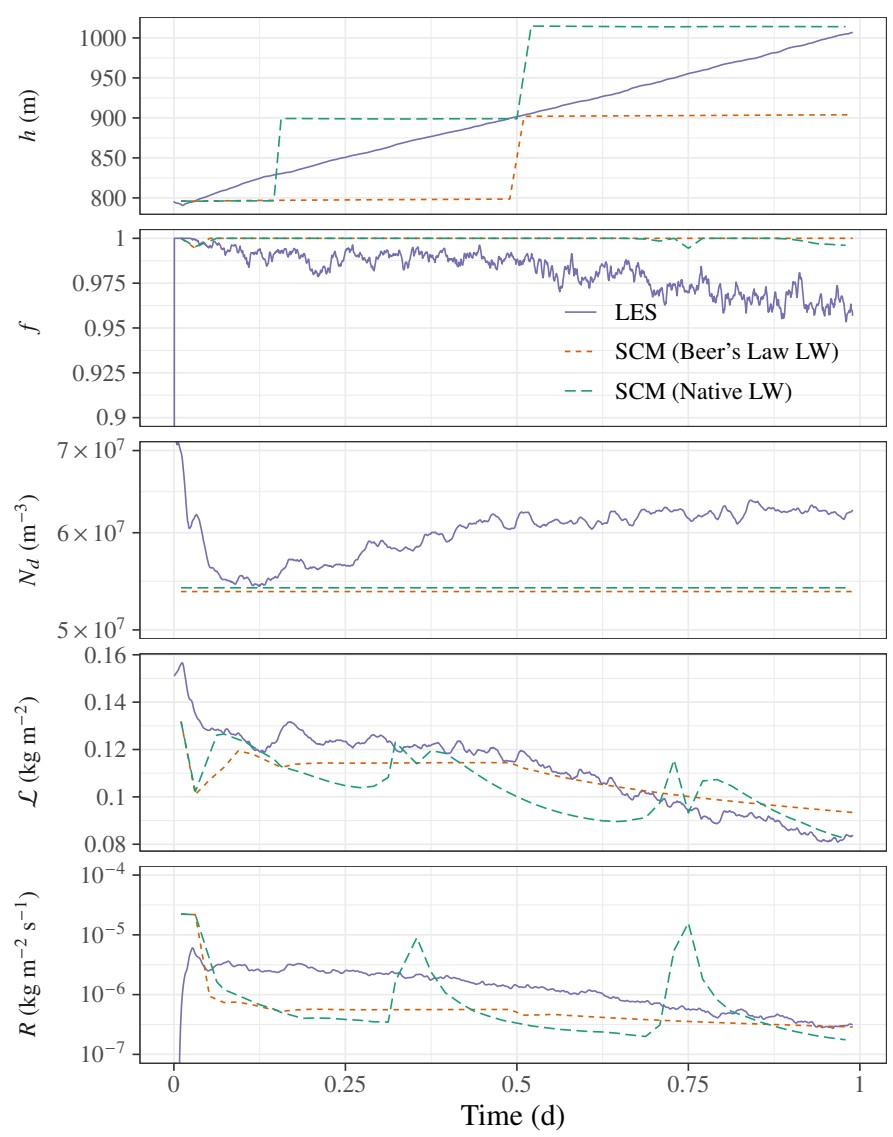

**Figure A1.** Evolution of domain-mean scalar diagnostics during 24-h simulations from DHARMA LES (solid blue line) and ModelE3 SCM, using Beer's Law parameterization of longwave raditive cooling per Ackerman et al. (2009) (red dotted line) and the ModelE3 native LW radiative transfer (green dashed line). The panels from the top depict domain-mean inversion height (location of maximum gradient in potential temperature below 5 km altitude), stratiform cloud cover (fraction of columns with opacity of at least 2.5 in the LES), cloud droplet concentration (average weighted by cloud water mixing ratio), liquid water path, and surface precipitation rate.

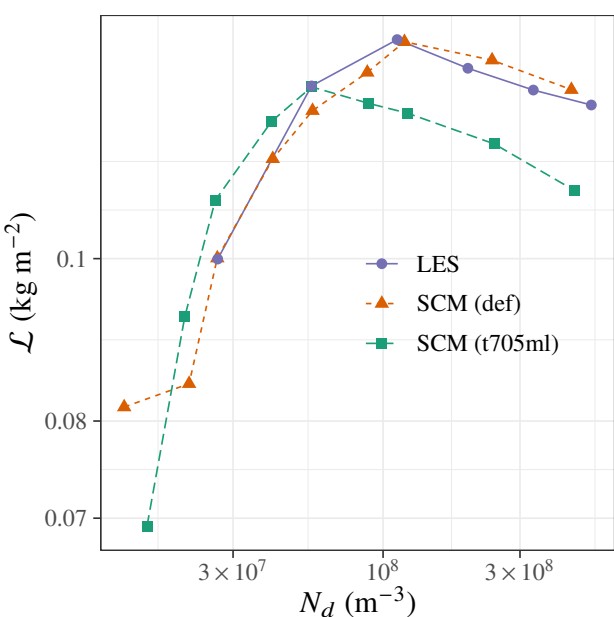

**Figure A2.** Domain mean $\mathcal{L}$ versus $N_d$ (vertical average weighted by cloud water mixing ratio) for the DHARMA LES (solid blue line) and two ModelE3 SCM configurations: the default tuning (red dotted line) analyzed in this study and the machine-learning tuning Tun1 (dashed green) used in Part 1. The outputs are averaged over hours 2–12, and Beer's Law parameterization of longwave flux divergence is used for the LES and SCM.