# Peer review of "Can GCMs represent cloud liquid water path adjustments to aerosol-cloud interactions?"

_EGUsphere, 2024_

## Referee Comment (RC2)

**Review of "Can GCMs represent cloud adjustments to aerosol–cloud interactions?" by Mülmenstädt et al. (egusphere-2024-778)**

Process-level modeling predicts a decrease in cloud water with aerosol through enhanced entrainment. However, these so-called negative cloud water adjustments are not captured in general circulation models (GCMs). To solve this apparent contradiction, this study presents and discusses a set of single-column model (SCM) simulations that use GCM model physics, as well as corresponding three-dimensional GCM simulations. Most interestingly, the presented SCM simulations show negative cloud water adjustments, which are missing in the GCM simulations. As the model physics are the same in SCMs and GCMs, the authors conclude that the absence of negative cloud water adjustments can result from internal feedbacks in the climate system. All in all, this is an exciting study that fits the scope of Atmospheric Chemistry and Physics. While I do not have any substantial concerns, I would like to make some suggestions below that the authors may want to address in a revised version.

**Minor Comments**

Ll. 22, 323 – 326.: To understand the integrated effect of clouds on the radiation budget, it makes no sense to express $RA_L$ as a function of $N_d$. However, to increase process-level understanding, separating $RA_L$ for low and high $N_d$ might be helpful.

Ll. 73 – 74: I suggest briefly summarizing the criteria of Medeiros and Stevens (2011).

Eq. 5: The term describing horizontal advection should also be multiplied by the air density.

Ll. 161 – 164: Why must two different notations be introduced to indicate mass-weighted vertical averages?

Eq. 7: The third term on the right-hand side of (7) might require a comment.

Ll. 179 – 181: Sub-meter turbulence plays an important role in the entrainment process by preconditioning the free-tropospheric air before entrainment, but it is the large-scale boundary layer circulation [O(1 km)] that drags the neutrally buoyant free-tropospheric air into the boundary layer (e.g., Yamaguchi and Randall 2012).

Ll. 219 – 222: A drier free troposphere (a more negative moisture jump) decreases the buoyancy jump, increasing entrainment. Thus, the displayed decrease in entrainment with the moisture jump is probably only due to the mentioned correlation with the temperature jump.

Ll. 300 – 302: $d\ln(L)/d\ln(N_d)$ is slightly negative for $Q_{sed} > 0$. Are these values negligible?

Ll. 320 – 321: Shortwave radiative heating should be saturated for sufficiently large L. How large can L be to be affected by this effect?

Ll. 333 – 335: Introduce the "three Sc regions" here?

Ll. 447 – 448: Should "subadiabaticity" be an additional model variable? Please clarify.

**Technical Comments**

Title: "cloud adjustments" to "cloud water adjustments"

L. 125: "$u^*$" to "$u_*$"

Fig. 1: Display y-axis in meters?

L. 307: "increases" to "decreases"?

**References**

Yamaguchi, T., & Randall, D. A. (2012). Cooling of entrained parcels in a large-eddy simulation. *Journal of the Atmospheric Sciences*, *69*(3), 1118-1136.

---

## Referee Comment (RC3)

**Review of „Can GCMs represent cloud adjustments to aerosol–cloud interactions?"**
**by Mülmenstädt et al.**

The manuscript presents some thought provoking anlysis on the disrepancy of short-term, limited area qunatifications of the liquid water path (LWP) adjustment to changes in anthropogenic aerosol. The authors show that GCMs that simulate a negative LWP adjustment in response to entrainment drying in a SCM case study, and when considering interannual variability in the present-day climate, quantify a non-existant or even slightly positive LWP adjustment when considering differences between the present-day and preindustrial climatological state.
The manusript fits well into the scope of ACP and provides a novel perspective, which will be of interest for a broad community. Some statements made by the authors are not sufficiently backed up by their analysis in my opinion (see comment below). Either further proof is required, or made statements will have to be softened considerably prior publication.

My main comment is with respect to your assertion, that the simulated mixing is free of numerical artifacts as you define them. Your shown associations to large-scale parameters and comparisons against LES in SCM, are a necessary, but by no means sufficient condition to support this statement. You still could get the right answer for the wrong reasons. Your analysis does not prove that you get the right answer for the right reasons.
One such proof could be a tracer mixing analysis in your SCM setup. If you had a tracer field above the BL, how consistent is your accumulation of tracer within the BL with your expected terms (given the very minor contribution of R, you can predict E given the initial conditions and large-scale forcing) and how is it affected by changes in vertical resolution? Such experiments would at least demonstrate the impact of numerical diffusion due the involved sharp gradients.
Secondly, equations 3-5 neglect the contribution of the convective mass flux. How sensitive are your SCM results to the shallow convection parameterisation and what percentage of grid points in the analysed GCM regions is impacted by the convection scheme? If that percentage is low, than you may be ok with neglecting this term.
Finally, the timescales of entrainment and deepening in the SCM ModelE3 analysis indicate a discrepancy with the timescales of LWP adjustment through entrainment and boundary layer deepening. These have been shown to be longer than a day, not within the first 6h. Thus the entrainment simulated, is still not in line with the physical mechanism. Furthermore, there does not seem to be a relation between the timing of the level jump in h and the time-integrated diagnosed Eq between the different model experiments. Combined with the fact that the vertical level jump occurs in the Na=120cm-3 run before the Na=160m-3, does suggest that the parameterised entrainment is not entirely physical.

Edits/Clarifications:

- L161 why A with caret and angular brackets to denote the same thing?
- Equ.7: what is $h^-$? Typo?
- L166: How are $E_{theta}$ and $E_q$ computed? By rearranging equs. 3-5 using equ. 7 for the material derivative? Please clarify.
- L185: Please rephrase „...., that is, lead to numerical diffusion"
- L187: what is a host of a perturbation? I suggest to rephrase.
- L201: Take sentence („In an Eulerian model,...") out of bracket. Its stand-alone and of equal importance to the previous statement made.
- L204: I assume you mean the convective mass flux here specifically? Please clarify.
- L364: Please rephrase, at the moment it reads like you have to adjust the filtering to the signal you want. In my oppinion the conclusion is, that NEP and SEP do not show a

response. This is only found in completely cloud-covered regions. Follow up question: Do you have a reason for why this is the case?

- L375-385: I agree with the first part of the argument presented in this paragraph. That E does not continue to increase as LWP decreases in response to the Nd increase. Eventually it decreases again in this self-limiting manner as the authors discuss.
However, the link to a further increase in Nd is not clear to me. What is the evidence that Nd usually continues to increase during the LWP adjustment? Timescale analyses of ACI seem to suggest the opposite: quick microphysical response and change in Nd, longer manifestation of LWP. It is does not clear to me, how the decrease in E with increasing Nd is of relevance for a self-regulating process.
- L 376: This should not be in brackets. Its just as likely as the other hypothesis. Your analysis does not provide a proof in one or the other direction.
- 379: Comment: „increased entrainment leads to loss of LWP". This is under the assumption that the cloud dynamics and morphology do not change. As the BL deepens, stronger updrafts are needed to maintain the coupling with deeper cloud cores and higher LWP.
- L388: „In terms of mechanistic..." Does this statement refer to RA or the Nd-L PD relationship? If the first, I agree, if the latter, where is the evidence?
- L401: comment on: „the buffering mechanism is that enhanced entrainment leads to sufficient liquid-water loss to shut off entrainment driven by cloud-top radiative cooling, protecting the clouds from further liquid loss" Changes in cloud dynamics as the BL deepens through entrainment may also contribute
- L404/405: Please rephrase and clarify „First, negative relationships..." This statement applies to in GCMs. You do not provide evidence that this is the case in opportunistic experiments (in and outside (!) the sub tropics) or LES. To my understanding you provide evidence, that limited short-term studies on limited domain, or isolated features, may not provide the entire picture. I.e. other processes may be at play that buffer the initial response.
- Fig.4: Please add the observed range of values to the simulated range to obtain a feeling of the realism of the simulations right away
- Fig. 5: I would argue that the E3SM SCM simulations are unsuitable to address the question at hand, since you simulate a completely different cloud evolution than observed.

---

## Author Comment (AC1)

**Response to RC1**

Mülmenstädt et al. study how changes in cloud-condensation nuclei (CCN) aerosols lead to adjustments in the liquid water path (LWP) of warm liquid clouds. The two prevailing hypotheses are that more CCN lead to stronger entrainment drying and precipitation suppression, which act to decrease and increase the LWP, respectively. The authors perform global-climate model (GCM) and single-column model experiments and analyze boundary layer heat and moisture budgets to identify causal relationships related to the proposed mechanisms. The results indicate that both proposed mechanisms are active in GCMs, but the enhanced-entrainment mechanism has a negligible effect on global-mean LWP. The authors conclude by interpreting these results and posing guiding questions for future research.

I believe that this topic is highly relevant to the aerosol-cloud-climate community, the analysis is well done, and the paper is clearly written. I have very little to say regarding criticisms of the current manuscript, but I offer a few ideas for additional analysis and discussion that I think could improve the paper. If the authors deem that these additional pieces would be beyond the scope of the study, then I believe the paper would also be publishable without them. I therefore recommend minor revision.

*We thank the reviewer for the encouraging remarks, engaging suggestions for further analysis, and corrections to the text. Our responses are inline below.*

**General Comments**

The authors conclude with a substantial discussion section in which they pose six guiding questions for future research. This is helpful for the community to think about next steps. However, it would be even more helpful if the authors could explicitly connect the questions to concrete examples from their analysis. For example, one guiding question is "What complexity is required (to simulate the global LWP adjustment)?" The authors suggest looking for the minimal set of parameterizations that capture relevant process understanding. Can the authors perform single-column experiments with a range of complexity to give some guidance about what this minimal set of parameterizations might look like in practice? The authors also pose the question "how representative are susceptibilities in small ensembles of individual cases?" Can the authors identify any cases with the single-column model in which the LWP adjustment differs substantially from the canonical LES cases that are widely studied? Where do we need to look to find this differing behavior? If the authors can provide specific, concrete examples from their analysis to motivate the six questions in the discussion, then I think the discussion would be more useful to the community.

*These are excellent suggestions for further study; hence why they are in the recommendations/road map section. We wish we already had the answers the reviewer is looking for, but we can only do so much at a time.*

If the enhanced-entrainment mechanism is in fact negligible for the global-mean LWP adjustment, as suggested by the results in the study, then what are the implications for the historical effective radiative forcing from aerosol-cloud interactions (ERFaci)? The enhanced-entrainment mechanism was used to justify a positive radiative adjustment from LWP changes in the Bellouin et al. (2020). If this positive radiative LWP adjustment is in fact negligible, then doesn't that imply an even stronger negative ERFaci? There is already a tension between ERFaci estimates derived from process understanding and aerosol-cloud relationships ("bottom-up estimates") and ERFaci estimates from global energy-budget constraints ("top-down estimates"), with the former predicting a stronger negative ERFaci. Do the current findings exacerbate this tension? How do we interpret this, and how do we more forward? Given that the paper concludes with a substantial forward-looking discussion, I was hoping that the authors would have discussed this topic.

*The 68% $RA_\mathcal{L}$ range of Bellouin et al. (2020) includes small positive values, so a small overall entrainment ACI mechanism is not necessarily the end of the world. However, the tension between the historical temperature and the process constraints is a good question to raise. Bellouin et al. (2020) derive a stronger constraint from the historical temperature than from process understanding. Unfortunately, the energy-budget constraint does not narrow down which components are causing the tension. This is a troubling blind spot, but addressing that problem would require at least as many recommendations as the $RA_\mathcal{L}$-focused list we already have!*

**Specific Comments**

- Line 44: I suggest changing "cloud the..." to "complicate the..." or "contradict the..." to avoid confusion because the noun form of "cloud" is used often in the preceding text.
  *There goes the first author's attempt at word play; changed.*

- Line 122: "The idealizations active in the baseline experiment are:" A colon should not be used here because colons should follow complete sentences, not sentence fragments (apologies for my obsession with grammar)
  *Thanks for the correction; changed.*

- Line 134: "top-of-atmosphere flux" → "top-of-atmosphere radiative flux"
  *Thanks for the correction; changed.*

- Line 368: consider changing "is small enough to require far longer model runs" to "is small enough to require far longer model runs to detect"
  *Thanks for the suggestion; changed.*

**Response to RC2**

**Review of "Can GCMs represent cloud adjustments to aerosol–cloud interactions?" by Mülmenstädt et al. (egusphere-2024-778)**

Process-level modeling predicts a decrease in cloud water with aerosol through enhanced entrainment. However, these so-called negative cloud water adjustments are not captured in general circulation models (GCMs). To solve this apparent contradiction, this study presents and discusses a set of single-column model (SCM) simulations that use GCM model physics, as well as corresponding three-dimensional GCM simulations. Most interestingly, the presented SCM simulations show negative cloud water adjustments, which are missing in the GCM simulations. As the model physics are the same in SCMs and GCMs, the authors conclude that the absence of negative cloud water adjustments can result from internal feedbacks in the climate system. All in all, this is an exciting study that fits the scope of Atmospheric Chemistry and Physics. While I do not have any substantial concerns, I would like to make some suggestions below that the authors may want to address in a revised version.

*We thank the reviewer for the encouraging remarks and improvement suggestions. Our responses are inline below.*

**Minor Comments**

- Ll. 22, 323 – 326.: To understand the integrated effect of clouds on the radiation budget, it makes no sense to express $RA_{\mathcal{L}}$ as a function of $N_d$. However, to increase process-level understanding, separating $RA_{\mathcal{L}}$ for low and high $N_d$ might be helpful.
  *Agreed that it would be a good technique for understanding processes. In practice, in this model, the fraction of the $N_d$ population in the rising branch of the inverted v is quite small, a result we showed in part 1, so we would hit diminishing returns. (This is one of the surprising changes since the CMIP5/AeroCom IND3 era.)*

- Ll. 73 – 74: I suggest briefly summarizing the criteria of Medeiros and Stevens (2011).
  *Thanks for the suggestion; added.*

- Eq. 5: The term describing horizontal advection should also be multiplied by the air density.
  *Thanks for the correction; fixed.*

- Ll. 161 – 164: Why must two different notations be introduced to indicate mass-weighted vertical averages?
  *Bad choice on the first author's part; fixed.*

- Eq. 7: The third term on the right-hand side of (7) might require a comment.
  *Thanks for the suggestion; added.*

- Ll. 179 – 181: Sub-meter turbulence plays an important role in the entrainment process by preconditioning the free-tropospheric air before entrainment, but it is the large-scale boundary layer circulation [O(1 km)] that drags the neutrally buoyant free-tropospheric air into the boundary layer (e.g., Yamaguchi and Randall 2012).
  *Indeed, and this is something km-scale models might do better than GCMs. In the revised manuscript, we have added this to the required-resolution/required-complexity part of the discussion section (as in, is bulk entrainment may be "good enough" for the climate response, or is resolving the mesoscale circulation essential?)*

- Ll. 219 – 222: A drier free troposphere (a more negative moisture jump) decreases the buoyancy jump, increasing entrainment. Thus, the displayed decrease in entrainment with the moisture jump is probably only due to the mentioned correlation with the temperature jump.
  *Agreed; this is what we meant by "(and the moisture jump being strongly correlated with the temperature jump)". We have made this clearer in the revised manuscript.*

- Ll. 300 – 302: dln(L)/dln(Nd) is slightly negative for Qsed > 0. Are these values negligible?
  *In the revised manuscript, we have clarified that increasingly negative susceptibilities as $Q_{sed}$ becomes more positive are the expected effect if there is an entrainment–sedimentation ACI mechanism active in the model.*

- Ll. 320 – 321: Shortwave radiative heating should be saturated for sufficiently large L. How large can L be to be affected by this effect?
  *It would be interesting to plot this as a function of $\mathcal{L}$, $N_d$, and the $N_d$ perturbation, and perform a budget analysis as in Hoffmann et al. (2020), but that is much more elaborate than we wanted for this simple sensitivity test.*

- Ll. 333 – 335: Introduce the "three Sc regions" here?
  *We're reluctant to interrupt the logical flow of this paragraph with a description of the regional selection. However, we have added a reference to the subsection of the Methods section describing this selection*

- Ll. 447 – 448: Should "subadiabaticity" be an additional model variable? Please clarify.
  *In the models we're using, subadiabaticity is an emergent property of the model. We've clarified in the text that it is to be diagnosed from existing variables.*

**Technical Comments**

- Title: "cloud adjustments" to "cloud water adjustments"
  *Agreed and changed (to "cloud liquid water path adjustments").*

- L. 125: "$u^*$" to "$u_*$"
  *Thanks for spotting. Corrected.*

- Fig. 1: Display y-axis in meters?
  *This is a bit difficult in practice. The model levels are not uniformly spaced in height, and since the model natively uses a hybrid sigma coordinate, the spacing also varies with surface pressure. Converting to geometric height would thus make the vertical gradients appear less sharp than the model physics sees them. We have quoted the average model level thickness in the captions instead.*

- L. 307: "increases" to "decreases"?
  *The first author managed to get himself confused with this negative quantity. Thank you for spotting the error. We have changed it to "makes $\partial \log \mathcal{L}/\partial \log N_d$ more negative".*

**References**

Yamaguchi, T., & Randall, D. A. (2012). Cooling of entrained parcels in a large-eddy simulation. Journal of the Atmospheric Sciences, 69(3), 1118-1136.

**Response to RC3**

**Review of „Can GCMs represent cloud adjustments to aerosol–cloud interactions?" by Mül-menstädt et al.**

The manuscript presents some thought provoking anlysis on the disrepancy of short-term, limited area qunatifications of the liquid water path (LWP) adjustment to changes in anthropogenic aerosol. The authors show that GCMs that simulate a negative LWP adjustment in response to entrainment drying in a SCM case study, and when considering interannual variability in the present-day climate, quantify a non-existant or even slightly positive LWP adjustment when considering differences between the present-day and preindustrial climatological state.

The manusript fits well into the scope of ACP and provides a novel perspective, which will be of interest for a broad community. Some statements made by the authors are not sufficiently backed up by their analysis in my opinion (see comment below). Either further proof is required, or made statements will have to be softened considerably prior publication.

*We thank the reviewer for the thorough reading and insightful comments. Responses inline below.*

My main comment is with respect to your assertion, that the simulated mixing is free of numerical artifacts as you define them. Your shown associations to large-scale parameters and comparisons against LES in SCM, are a necessary, but by no means sufficient condition to support this statement. You still could get the right answer for the wrong reasons. Your analysis does not prove that you get the right answer for the right reasons.

One such proof could be a tracer mixing analysis in your SCM setup. If you had a tracer field above the BL, how consistent is your accumulation of tracer within the BL with your expected terms (given the very minor contribution of R, you can predict E given the initial conditions and large- scale forcing) and how is it affected by changes in vertical resolution? Such experiments would at least demonstrate the impact of numerical diffusion due the involved sharp gradients.

*There are many good suggestions here for further work. We have clarified in the text that we are comparing to the foundational process understanding on which the enhanced-entrainment picture is based (Ackerman et al., 2004; Bretherton et al., 2007), neither of which include a tracer analysis, and that consistency with those LES studies does not guarantee the absence of other artifacts not specifically searched for.*

Secondly, equations 3-5 neglect the contribution of the convective mass flux. How sensitive are your SCM results to the shallow convection parameterisation and what percentage of grid points in the analysed GCM regions is impacted by the convection scheme? If that percentage is low, than you may be ok with neglecting this term.

*Thank you for raising this important point. We have noted in the revised Methods section that nonlocal transport through the PBL top by parameterized convection is not included in the PBL budget-based entrainment diagnostics. We have also clarified that the convection scheme does not trigger in either model in the SCM run. We have further clarified that shallow convection and cloud macrophysics are unified in E3SM and do not produce nonlocal vertical transport. Finally, we have added that we discard time steps where "deep" convection fires in the 3D atmosphere run, which occurs in 2% of time steps and which we forgot to note in the original manusript.*

Finally, the timescales of entrainment and deepening in the SCM ModelE3 analysis indicate a discrepancy with the timescales of LWP adjustment through entrainment and boundary layer deepening. These have been shown to be longer than a day, not within the first 6h. Thus the entrainment simulated, is still not in line with the physical mechanism. Furthermore, there does not seem to be a relation between the timing of the level jump in h and the time-integrated diagnosed Eq between the different model experiments. Combined with the fact that the vertical level jump occurs in the Na=120cm-3 run before the Na=160m-3, does suggest that the parameterised entrainment is not entirely physical.

*The ModelE3 SCM closely matches the behavior of the DYCOMS-II LES (see appendix). The DYCOMS-II LES, in turn, is one of the foundations of our process understanding of the entrainment ACI mechanism (Ackerman et al., 2004, 2009). The ACI adjustment timescales certainly cover a spectrum, and there are situations where the deepening is slower (or faster) than in the DYCOMS-II RF02 case, but we disagree with the reviewer's claim that there is a discrepancy between the ModelE3 SCM and process understanding in the case presented here.*

**Edits/Clarifications:**

- L161 why A with caret and angular brackets to denote the same thing?
  *This was a bad choice of notation. We have fixed it in the revised manuscript.*

- Equ.7: what is $h^-$? Typo?
  *$A|z = h^-$ means A is to be evaluated just below the inversion, which we forgot to explain in the text. We have added the explanation to the revised manuscript.*

- L166: How are Etheta and Eq computed? By rearranging equs. 3-5 using equ. 7 for the material derivative? Please clarify.
  *We have clarified that in the revised manuscript that each of the budget equations can be solved for the corresponding entrainment flux.*

- L185: Please rephrase „...., that is, lead to numerical diffusion"
  *Thank you for spotting this long run-on sentence. We have broken it up into shorter sentences.*

- L187: what is a host of a perturbation? I suggest to rephrase.
  *Thank you for pointing out where we could make the language more accessible. We have changed "host" to "multitude".*

- L201: Take sentence („In an Eulerian model,...") out of bracket. Its stand-alone and of equal importance to the previous statement made.
  *We disagree that parentheses indicate lower importance. These sentences are in parentheses because they are separate from the logical flow of the paragraph.*

- L204: I assume you mean the convective mass flux here specifically? Please clarify.
  *We have specified in the revised manuscript that we mean entrainment mass flux.*

- L364: Please rephrase, at the moment it reads like you have to adjust the filtering to the signal you want. In my oppinion the conclusion is, that NEP and SEP do not show a response. This is only found in completely cloud-covered regions. Follow up question: Do you have a reason for why this is the case?
  *This paragraph already contained multiple statements of caution. We describe the behavior as "hints", list the various caveats, and conclude with "On balance, the conservative interpretation of these results is that any potential $\mathcal{L}$ reduction signal in response to anthropogenic aerosol is small enough to require far longer model runs to detect." In the revised manuscript, we have further changed "However, the results come with multiple caveats" to "However, there are multiple reasons these results should be treated with caution until they can be confirmed in longer model runs".*

  *These results are included in the manuscript because we found it interesting that a subset of the columns (approximately 1/3 of columns selected for the default analysis have cloud fraction exactly 1) behaved in such close agreement with the canonical understanding of the effect of free-tropospheric relative humidity on entrainment drying. We felt it was more appropriate to report the results, fully caveated, than to undiscover them.*

  *As to the follow-up question, we would have to speculate. It is possible that it is simply a statistical fluctuation (which may be the reviewer's concern), hence our remark that longer model runs would be needed for confirmation. It is also possible that entrainment signals consistent with our physical understanding emerge most strongly when the grid box is fully cloudy, reducing the effect of the clear-sky fraction of the column. (This is part of the motivation for the $f > 0.9$ requirement, as well.)*

- L375-385: I agree with the first part of the argument presented in this paragraph. That E does not continue to increase as LWP decreases in response to the Nd increase. Eventually it decreases again in this self-limiting manner as the authors discuss. However, the link to a further increase in Nd is not clear to me. What is the evidence that Nd usually continues to increase during the LWP adjustment? Timescale analyses of ACI seem to suggest the opposite: quick microphysical response and change in Nd, longer manifestation of LWP. It is does

not clear to me, how the decrease in E with increasing Nd is of relevance for a self-regulating process.

*We agree with everything the reviewer says and note that the discussion of the evolution of a single cloud field involves only $\mathcal{L}$, in agreement with the reviewer. The discussion of $N_d$ is in the ensemble-average ("cloud aggregate statistics" in the manuscript) sense; at higher $N_d$, the cloud is more likely to enter a low-$\mathcal{L}$, nonentraining state because higher-$N_d$ clouds, on average, have lower $\mathcal{L}$.*

- L 376: This should not be in brackets. Its just as likely as the other hypothesis. Your analysis does not provide a proof in one or the other direction.

  *The reviewer appears to be of the opinion that parentheses indicate lesser importance. This is not the intent here, as is made clear by "we reiterate", "as throughout", and should also be clear from the context that part 1 of this manuscript series was devoted to exposing the importance of confounding. The parentheses are merely meant to indicate to the reader that this sentence is not part of the logical progression of the paragraph but points into a different, equally valid, direction.*

- 379: Comment: „increased entrainment leads to loss of LWP". This is under the assumption that the cloud dynamics and morphology do not change. As the BL deepens, stronger updrafts are needed to maintain the coupling with deeper cloud cores and higher LWP.

  *Thank you for this comment; we have noted that this explanation applies as long as the cloud remains surface-coupled closed-cell Sc.*

- L388: „In terms of mechanistic..." Does this statement refer to RA or the Nd-L PD relationship? If the first, I agree, if the latter, where is the evidence?

  *This paragraph summarizes the conclusion of a published paper, but we are happy to entertain the reviewer's intriguing point. We note that the claim is not that confounding is the only explanation, but the "simplest". Evidence for confounding mechanisms (e.g., by precipitation) is presented in the reference. The alternative explanation is that the negative correlation between $N_d$ and $\mathcal{L}$ in PD internal variability is indeed indicative of a causal relationship, but that this causal relationship is no longer manifest when taking the ensemble average of clouds whose $N_d$ has been systematically increased. There may be a mechanism that can explain such behavior, but it must be much less "simple" than the ubiquitous mechanisms of correlations due to confounding.*

- L401: comment on: „the buffering mechanism is that enhanced entrainment leads to sufficient liquid-water loss to shut off entrainment driven by cloud-top radiative cooling, protecting the clouds from further liquid loss" Changes in cloud dynamics as the BL deepens through entrainment may also contribute

  *That is a good point. It would be difficult to test the cloud dynamics response mechanism in the GCM, so we have changed the sentence to "a candidate for the buffering mechanism is that enhanced entrainment leads to sufficient liquid-water loss . . . ".*

- L404/405: Please rephrase and clarify „First, negative relationships..." This statement applies to in GCMs. You do not provide evidence that this is the case in opportunistic experiments (in and outside (!) the sub tropics) or LES. To my understanding you provide evidence, that limited short-term studies on limited domain, or isolated features, may not provide the entire picture. I.e. other processes may be at play that buffer the initial response.

  *We have added "in GCMs" to the sentence. We agree with the reviewer's interpretation; the value provided by GCMs is not a "true number" for ERFaci, but rather the ability to crosscheck the short-term, limited-domain process modeling evidence; the episodic, isolated, and often strongly perturbed opportunistic experiments evidence; and the difficult-to-disentangle questions of causality in satellite correlation studies.*

- Fig.4: Please add the observed range of values to the simulated range to obtain a feeling of the realism of the simulations right away

  *The case specification is called DYCOMS-II "research flight" 02, but in actuality it is an idealized case that is based on observations from that flight, averaging over multiple cloud types (Ackerman et al., 2009). The causal $N_a$ perturbation experiments are not possible to conduct in observations. Thus, the real test is whether the SCM reproduces the results from the LES studies for which the case specification was designed.*

- Fig. 5: I would argue that the E3SM SCM simulations are unsuitable to address the question at hand, since you simulate a completely different cloud evolution than observed.

  *First, as mentioned above, "observed" is ill-defined for this idealized case. Second, the behavior is not "completely different" but rather differs mainly in the precipitation state of the model, a longstanding and well known problem area for GCM physics. Third, different modeling centers have different tuning strategies. For GISS, SCM comparisons with a number of LES cases was used for parameterization development and for constraining uncertain coefficients used for GCM tuning. For E3SM, individual cases are studied in standalone papers (like this one), but agreement with cases is not part of the tuning strategy. Thus, expecting close agreement with DYCOMS-II LES from E3SM would be unrealistic. Even the fact that E3SM produces $\partial \log \mathcal{L}/\partial \log N_d$ of the correct sign is surprising; that is the main point of the paper, as mentioned several times in the text.*

**Response to RC4**

**Summary**

This manuscript is a timely investigation of the ability of GCMs and their underlying single-column models to qualitatively capture adjustments of liquid water path in stratocumulus clouds (Sc) to an increase in aerosol, relative to the behavior of liquid water path in response to aerosol changes known from large eddy simulations (LES). The question is addressed whether GCMs are incorrect due to parametric or base-state representation errors, or whether relatively few canonical cases simulated with LES are sufficiently representative to serve as a benchmark for the behavior of Sc liquid water path in response to aerosol changes as simulated by GCMs globally.

The first interesting insight provided by this work is that SCMs are qualitatively able to represent, via the mechanisms known from LES (precipitation suppression and entrainment drying), the liquid water path adjustments.

The second interesting insight provided is that GCMs can produce different liquid water path adjustments relative to canonical LES cases, possibly for the right reasons such as environmental variability, and possibly due to limitations such as limited model resolution, uncertainty in process representations, and uncertainty in the representation of the atmospheric base state.

A set of approaches to improve the ability of GCMs to represent the liquid water path adjustment to aerosol changes is presented and discussed, a valuable contribution to advance the science.

The section explaining the entrainment diagnostics is very nicely written. Other parts of the manuscript would benefit greatly from a similarly linear and didactic writing style, and certainly from much shorter sentences.

This reviewer recommends this manuscript for publication after minor revisions.

*We thank the reviewer for the encouraging remarks and suggestions for improving the manuscript. Responses inline below.*

**Specific Points**

Title: "Can GCMs represent cloud adjustments to aerosol–cloud interactions?"

The title is a bit misleading in that it overstates the scope of this work. After all, only the adjustment of liquid water path to aerosol-cloud interactions is investigated, and for a very limited subset of stratocumulus clouds.

Please change the title to something more representative, e.g.,

"Can GCMs represent the liquid water path adjustment to aerosol–cloud interactions?"

*Agreed and done.*

Section 2.1: Please specify that E3SM and ModelE3 are both used in SCM mode, but only E3SM is used in GCM mode.

*Done.*

Line 260: "We may be mitigating the E3SM artifact by averaging over two full deepening cycles, effectively averaging over the dependence of E on position in the deepening cycle. We attempt to mitigate the ModelE3 artifact by only averaging E until the first PBL deepening occurs."

Please justify in the text that this subsetting is not cherry-picking the results that produce a particular outcome.

*The reviewer is correct; the fact that we need to find workarounds for the artifacts in the entrainment diagnostics is far from satisfying. In the revised manuscript, we provide more rationale for why our choice of conditional averaging is a better representation of the model behavior than the unconditional-averaging alternative.*

Lines 277 to 281: "Whether the effect of varying the aerosol concentration on L is expected depends on our Bayesian prior. If our expectation for the L response is based on the RA L results of Mülmenstädt et al. (2024), we would predict the causal effect of increased Nd to be an increase in L."

The results of Mülmenstädt et al. (2024) represent the behavior of a cloud population simulated in that work, whereas the clouds simulated with SCMs in this work represent one particular set of conditions. Why is it valid to formulate an expectation based on the former and expect it to hold for the latter?

*Thank you for pointing out the problems with this argument. (We also note that the "expectation" and "surprises" are merely a narrative device for introducing the results of the study, so the results themselves are unaffected.) What we meant to say here is that the agreement on $RA_{\mathcal{L}} < 0$ across GCMs (Gryspeerdt et al., 2020) and the absence of enhanced entrainment physics in some GCMs (e.g., Salzmann et al., 2010) have led to the notion that GCMs may be structurally incapable of representing the enhanced entrainment mechanism known from LES (e.g., Zhou and Penner, 2017), while a long line of parameterization work (e.g., Randall et al., 1985; Lock et al., 2000; Bretherton and Park, 2008; Guo et al., 2011; Karset et al., 2020) indicates that it may well be possible to represent this mechanism in GCMs, either through direct parameterization or as an emergent behavior. We have reworded this section to make the tension between these interpretations clearer.*

"If our expectation is based on the LES-based process understanding, then we would predict the causal effect of Nd to be a decrease in L." ... "The surprising result is that the SCM sides with the LES, not the response of the 3D GCM with which the SCM shares its model physics."

Again, why is this surprising, given that the 3D GCM simulates a population of clouds whereas the SCM and LES runs simulate only one particular case? It is perfectly possible that the population of clouds behaves differently from the particular SCM case.

*The surprise here is specifically in the comparison with the 3D atmosphere run that shows no $\mathcal{L}$ reduction emerges when precipitation suppression disabled. The comment is still correct – we are comparing a single case against a climatology – but it is reasonable to expect to see at least some signal in the climatology if a representative Sc case (representative by design) shows evidence for the enhanced entrainment mechanism and if the opposing-sign mechanism has been disabled in the 3D model. We have clarified this slightly in the revised manuscript.*

Line 281: Please add a reference to Appendix A.

*Added.*

Line 332: "The L values at which entrainment turns on are in reasonable agreement with recent LES (Hoffmann et al., 2020) and observational (Zhang et al., 2022) results."

Too unspecific. Please detail how the agreement is reasonable with Hoffmann et al. (2020) and Zhang et al. (2022).

*First, we apologize for an error; we should have cited Zhang and Feingold (2023), not Zhang et al. (2022). The former explicitly give $50 \ g \ m^{-2}$ as the threshold (paragraph 3.2.3). Hoffmann et al. (2020) are less explicit, but, depending on choice of entrainment parameterization and for their $LWP_\infty = 60 \ g \ m^{-2}$ simulation, their Fig. 4 shows a sharp turn-on that saturates between approximately 20 and $40 \ g \ m^{-2}$. In the absence of a rigorous metric for agreement, we settled on "reasonable", but we have included this longer explanation in the revised manuscript.*

Line 336: "The main conclusion from these plots is that the model produces greater entrainment in response to higher Nd in Sc clouds with strong entrainment."

The response of entrainment to Nd is not shown as a function of entrainment. Please explain how one can draw the given conclusion.

*We have amended the revised manuscript to say "in Sc clouds with high enough $\mathcal{L}$ to support strong entrainment".*

Lines 348-356: This paragraph gives the impression of a stream of consciousness reflecting the authors' familiarity with their work. It is presented without references to figures or other supporting material, and invoking not-shown results, all of which makes it very hard to untangle. It is not obvious that it is needed, but if it is, it requires rewriting.

*On rereading this paragraph, we agree that it is an imposition on the reader and that it is not needed. We have removed it in the revised manuscript.*

Line 392: The evidence for this causal relationship comes from SCM studies, where, like Guo et al. (2011), we find that increased aerosol, while holding all other boundary conditions fixed, leads to liquid-water loss.

This is not true for all Nd values, is it? Please qualify this finding accordingly.

*Specified that this occurs at sufficiently high $N_d$.*

**References**

[revised manuscript text omitted]